# Double cross-linked 3D layered PBI proton exchange membranes for stable fuel cell performance above 200 °C

Liang Zhang[1], Mengjiao Liu[1], Danyi Zhu[1], Mingyuan Tang[1], Taizhong Zhu[1], Congjie Gao[1], Fei Huang ☉[1] ✉ & Lixin Xue ☉[2,3] ✉

Phosphoric acid doped proton exchange membranes often experience performance degradation above 200 °C due to membrane creeping and phosphoric acid evaporation, migration, dehydration, and condensation. To address these issues, here we present gel-state polybenzimidazole membranes with double cross-linked three-dimensional layered structures via a polyphosphoric acid sol-gel process, enabling stable operation above 200 °C. These membranes, featuring proton-conducting cross-linking phosphate bridges and branched polybenzimidazole networks, effectively anchor and retain phosphoric acid molecules, prevent 96% of its dehydration and condensation, improve creep resistance, and maintain excellent proton conductivity stability. The resulting membrane, with superior through-plane proton conductivity of 0.348 S cm$^{-1}$, delivers outstanding peak power densities ranging from 1.20–1.48 W cm$^{-2}$ in fuel cells operated at 200-240 °C and a low voltage decay rate of only 0.27 mV h$^{-1}$ over a 250-hour period at 220 °C, opening up possibilities for their direct integration with methanol steam reforming systems.

Fuel cells have garnered significant attention owing to their clean and high-efficiency energy conversion, among which proton exchange membrane fuel cells (PEMFCs) were particularly applicable for portable electronic and transportation devices due to their high output power and quick start-up time[1–3]. However, their development was still constrained by the challenges from the safe storage and transportation of high-purity hydrogen gas as well as the construction of safe hydrogen fueling infrastructure[4].

Operating PEMFCs above 200 °C to match the temperature requirements of methanol-reforming processes may allow the direct use of methanol as fuel to avoid the challenging issues from hydrogen storage, transportation, and refueling[5]. Integrated methanol steam reforming proton exchange membrane fuel cell systems (MSR-PEMFCs), directly supplying hydrogen-rich gas for fuel cells from methanol steam reformation without intermediate purification or

cooling steps, could also have enhanced energy conversion efficiency with simplified processes[5,6]. However, the current operational temperature range of conventional PEMFCs (60–160 °C) falls short of the higher reaction temperature demands (200–300 °C) of methanol steam reforming. To effectively operate an MSR-PEMFC power system, it is imperative to run PEMFCs at temperatures exceeding 200 °C, aligning with methanol reforming gas systems[6].

Phosphoric acid (PA)-doped polybenzimidazole (PBI) membranes using PA molecules as the medium to facilitate proton conduction above 120 °C[7] tends to lose performance above 160 °C due to creep deformation[8–10] and the loss of absorbed PA by evaporation, migration and dehydration[7,11]. Conventional cross-linking may reduce creeping and improve the mechanical stability of PBI-based proton exchange membranes (PEMs), but, at the same time, caused a reduction in PA binding sites and declination in proton conductivity[12]. For example, Na

[1]Center for Membrane Separation and Water Science & Technology, College of Chemical Engineering, Zhejiang University of Technology, 310014 Hangzhou, China. [2]College of Chemistry and Materials Engineering, Wenzhou University, 325035 Wenzhou, Zhejiang, China. [3]Institute of New Materials & Industrial Technologies, Wenzhou University, 325024 Wenzhou, China. ✉e-mail: feihuang@zjut.edu.cn; xuelx@zjut.edu.cn

et al. used 1,3-di(2,3-epoxypropoxy)−2,2-dimethylpropane as a bi-functional cross-linking agent to enhance the mechanical strength of the PBI membrane, resulting in decreased proton conductivity from 0.014 S cm$^{-1}$ to 0.009 S cm$^{-1}$ at 200 °C[13]. To address above trade-off issue, Wang et al. used a pillar[5]arene to prepare a proton exchange membrane with heterogeneous cross-linking arrangement, resulting in improved mechanical strength (14.6 MPa) and PA retention. The resulted PEM with a proton conductivity of 0.240 S cm$^{-1}$ at 180 °C, achieved minimal voltage decay of 0.04 mV h$^{-1}$ in fuel cell operated over a 200 h at 160 °C[14].

To avoid PA loss from PA-PBI PEMs under high temperatures, functional groups such as hydroxyl[15] or amino[16] groups may be introduced to strengthen the interaction between PA and PBI, or inorganic oxide fillers such as TiO$_2$[17], SiO$_2$[18], or ZrO$_2$[19] may be included to anchor PA in the polymeric networks. For example, Jiang et al.[18] incorporated phosphotungstic acid-impregnated mesoporous silica into the PA-PBI matrix to form PA-phosphosilicate nanoclusters and suppress PA leaching, achieving impressive fuel cell operating stability at 200 °C over 2700 h. However, the fuel cell power output was still poor (386 mW cm$^{-2}$ at 200 °C) when PA-PBI PEMs were prepared in dense forms from traditional organic solvent casting processes[18].

In this study, polyphosphoric acid (PPA) based sol–gel methodology[20] was implemented to generate gel-state PBI-G PEMs with high PA uptake and proton conductivity. HO- groups were built onto PBI networks to anchor PPA and PA molecules and yield proton-conducting cross-linking phosphate bridges while the addition of tri-functional trimesic acid (TMA) monomer bestowed double cross-linking among the formed branched PBI polymeric chains, leading to the formation of stable DC-PBI-G three-dimensional (3D) layered structures with higher proton conductivity and greatly reduced creeping above 200 °C. Double cross-linked PA-doped gel-state PBI PEMs (DC-PBI-G) with 3D layered structures, as well as their derivative

from further treatment under 240 °C (DC-PBI-G-240), were prepared from the condensation polymerization of 3,3'-diaminobenzidine (TAB), 2,5-dihydroxy-terephthalic acid (DHTA), and TMA via the PPA sol–gel process. The impact from such double cross-linking on the structure, chemical and mechanical stability, PA retention, proton conductivity and fuel cell performance of the formed DC-PBI-G PEMs was systematically studied in the temperature range of 200−240 °C in comparison with gel-state HO-PBI-G with single cross-linking involving phosphoric anchoring bridges, gel-state p-PBI-G without cross-linking, as well as dense state m-PBI membranes (m-PBI-D) prepared from traditional organic solvent casting method.

## Results

### Gel-state 3D layered microstructure

As shown in Fig. 1, double cross-linked DC-PBI-G and DC-PBI-G-240 were prepared via the PPA sol−gel methods, using PPA as the polymerization solvent and condensing agent for tetraamine and dicarboxylic acid monomers. In this study, tri-functional TMA was added as a branching agent, while HO-functionalized DHTA was used as dicarboxylic acid monomer, to fabricate double cross-linked PEMs, possessing the in-situ anchoring sites for PA and PPA chains or bridges attachment and cross-linking. The reaction was executed under controlled heating and terminated upon achieving moderate viscosity around 190 °C. The DC-PBI-G PEMs were obtained by casting the polymer solution onto glass surfaces, followed by undergoing a sol−gel phase inversion process. To fabricate the DC-PBI-G-240 PEMs, an additional step involved curing the DC-PBI-G PEMs at 240 °C for 10 h through an intensified in-situ PA anchoring approach.

The addition of branched monomers TMA in the PPA solutions tend to yield cross-linked PBI that is less soluble in PPA, resulting in precipitation, premature reaction termination, reduction in molecular weight, and ultimately decrease in mechanical properties. To avoid these issues, we opted to lower the total monomer concentration in

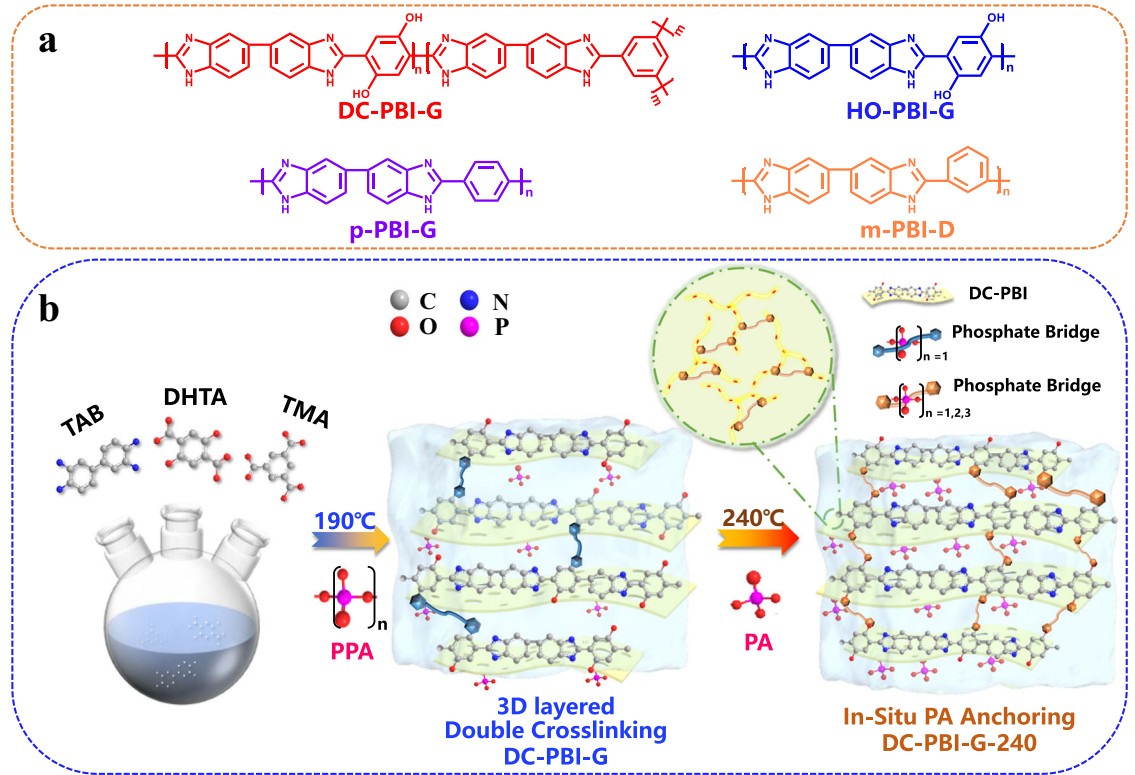

**Fig. 1 | Chemical structures and membrane preparation process. a** Chemical structure of PBI membranes involved in this study. **b** Preparation process of PA-doped gel-state DC-PBI-G and DC-PBI-G-240 membranes.

PPA to the lowest limit of 2 wt% for the PPA sol–gel process at present, keeping the TMA ratio less than 10% of the total carboxylic acid monomer content. For branched monomer contents ≥10%, PA-doped membrane formation was unattainable.

The branched monomer (TMA) concentration was varied to study the membrane-forming properties and mechanical properties of the processed membranes, as outlined in Table 1. A content of 9.1% allowed for membrane formation, but exhibits poorer mechanical properties (creep rate = $300.0 \times 10^{-12}$ Pa$^{-1}$ s$^{-1}$). It was noteworthy that the incorporation of branching in the monomer to a level of 4.8% resulted in a significant decrease in creep rate to $13.8 \times 10^{-12}$ Pa$^{-1}$ s$^{-1}$, as depicted by the creep compliance curve illustrated in Supplementary Fig. 1. Therefore, we prepared DC-PBI-G via the PPA sol–gel process, with a branched monomer TMA ratio of 4.8%.

As depicted in Fig. 2a, e, it was observed that dense m-PBI-D membranes from the traditional fabrication process exhibit a uniform and dense structure. The PPA sol–gel process allowed PBI polymeric chains to interact freely with PPA molecules in the sol state, resulting in the formation of favorable 3D layered microstructures in p-PBI-G, HO-PBI-G, and DC-PBI-G, as shown in Fig. 2 b–d, f–h with much higher

porosity, pore volume, and surface area (Supplementary Table 1). Furthermore, such 3D layered structures were well-preserved in DC-PBI-240 even after exposure to high-temperature treatment (Supplementary Fig. 2).

**Table 1 | Effects of TMA content on creep compliance and creep rate of membranes**

| PA-doped membrane | Monomer charge (wt%) | Branched monomer content (mol%)[a] | $J^0_s$ ($10^{-6}$ Pa$^{-1}$) | Creep rate ($10^{-12}$ Pa$^{-1}$ s$^{-1}$) |
|---|---|---|---|---|
| DC-PBI-G | 2 | 4.8 | 2.1 ± 0.1 | 13.8 ± 2 |
| | 2 | 9.1 | 4.1 ± 0.3 | 300.0 ± 8 |
| | 2 | 10.0 | N/A[b] | N/A[b] |
| | 2 | 28.6 | N/A[b] | N/A[b] |
| | 2 | 100.0 | N/A[b] | N/A[b] |

The "±" numbers represent the average of at least three testing samples.
[a]The molar content of the branched TMA monomer as a fraction ratio of the total molar content of carboxylic acid monomers.
[b]The formation of PA-doped membrane was unattainable via the PPA sol–gel process.

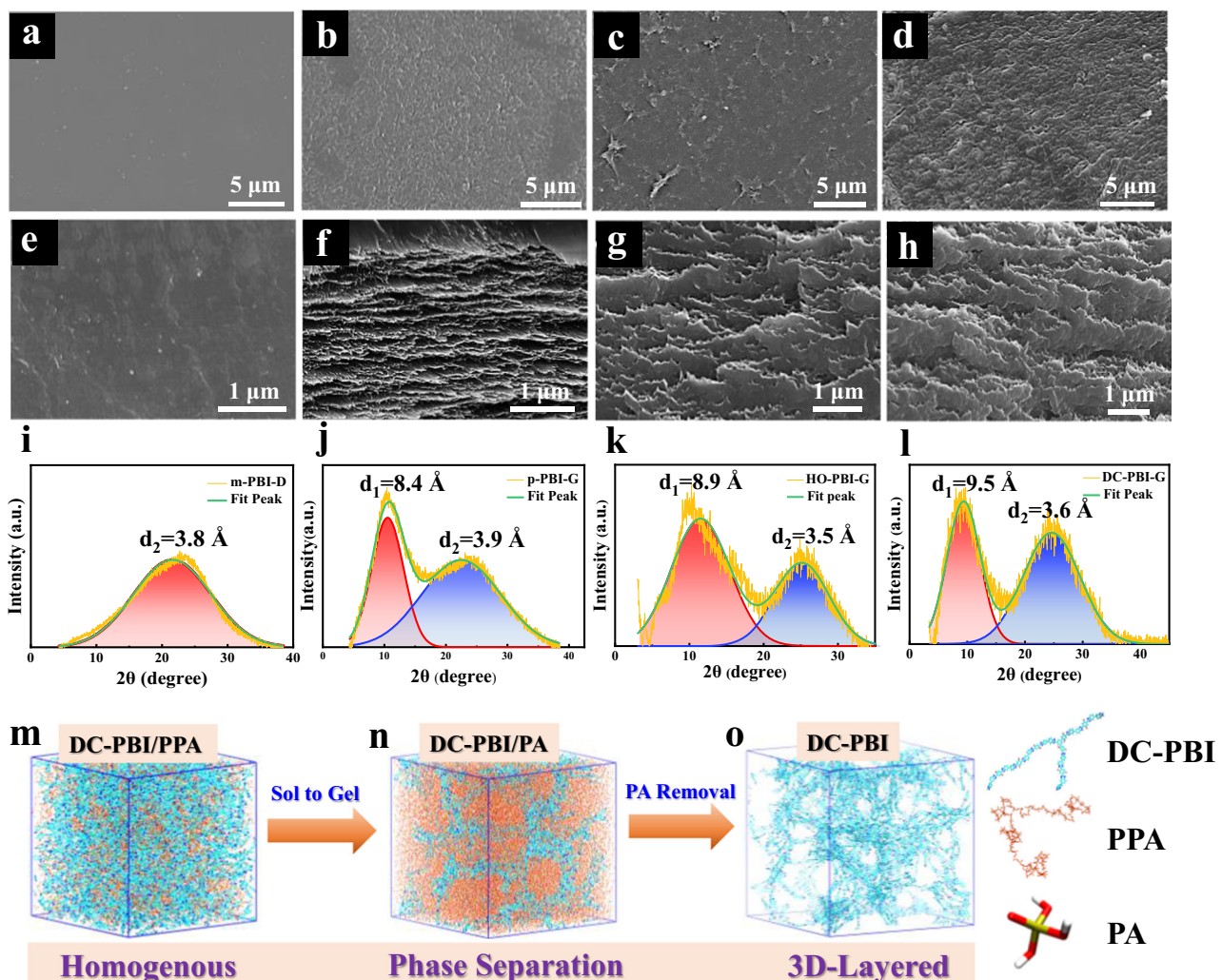

**Fig. 2 | Micromorphological characterizations and membrane formation mechanism.** SEM surface images of PA-undoped (**a**) m-PBI-D, (**b**) p-PBI-G, (**c**) HO-PBI-G, and (**d**) DC-PBI-G. SEM cross-sectional images of PA-undoped (**e**) m-PBI-D, (**f**) p-PBI-G, (**g**) HO-PBI-G, and (**h**) DC-PBI-G. WAXD spectra of PA-undoped (**i**) m-PBI-D, (**j**) p-PBI-G, (**k**) HO-PBI-G and (**l**) DC-PBI-G. 3D visualization and spatial configuration of the DC-PBI modeled structure within a $12 \times 12 \times 12$ nm$^3$ amorphous cubic cell, depicted in (**m**) the presence of PPA, (**n**) the presence of PA, and (**o**) post-PA removal.

**Table 2 | Polymer characteristics and PA-doped PBI membrane composition**

| PA-doped membrane | Polymer content (wt%) | PA content (wt%) | ADL$_T$ $^a$(PA PRU$^{-1}$) | IV(dL g$^{-1}$) | $M_w$ $^b$ (kDa) |
|---|---|---|---|---|---|
| m-PBI-D | 27.6 ± 2.3 | 56.9 ± 5 | 6.1 ± 1 | 1.8 | 93.8 |
| p-PBI-G | 3.2 ± 0.5 | 58.7 ± 6 | 59.3 ± 2 | 3.1 | 174.1 |
| HO-PBI-G | 3.2 ± 0.4 | 60.2 ± 6 | 63.0 ± 2 | N/A$^c$ | N/A$^c$ |
| DC-PBI-G | 3.1 ± 0.5 | 63.8 ± 5 | 66.7 ± 2 | N/A$^c$ | N/A$^c$ |
| DC-PBI-G-240 | 5.18 ± 0.4 | 91.5 ± 3 | 48.5 ± 3 | N/A$^c$ | N/A$^c$ |

The "±" numbers represent the average of at least three testing samples.
$^a$Obtained by acid-base titration using 0.1 M NaOH.
$^b$Estimated from the inherent viscosity.
$^c$The membrane was insoluble in 98 wt% $H_2SO_4$ and the values could not be tested.

WAXD analyses illustrated in Fig. 2i–l, showed that the dense m-PBI-D membrane displayed a single diffraction peak centered at 23.0°, which corresponds to a $d_2$ spacing of 3.8 Å due to the tight stacking of PBI chains. In contrast, the gel-state membranes with 3D layered structures exhibited two diffraction peaks (Supplementary Table 2), with $d_2$ spanning 3.6–3.9 Å, which was consistent with the value observed in m-PBI-D. Notably, an additional peak with $d_1$ spacing ranging from 8.4–9.5 Å was observed for all the membranes obtained from the unique PPA sol–gel process, characterizing the formation of the 3D layered microstructures in p-PBI-G, HO-PBI-G and DC-PBI-G. Cross-linking by phosphate bridges shrunken the $d_2$ packing spacing from 3.9 Å in p-PBI-G to 3.5 Å in HO-PBI-G, while expanding $d_1$ spacing from 8.4 Å in p-PBI-G to 8.9 Å in HO-PBI-G. Additional cross-linking caused by the branched PBI chains further expanded $d_1$ peak spacing to 9.5 Å in DC-PBI-G PEMs. Following the high-temperature treatment at 240 °C, a further enlargement in $d_1$ spacing of 10.2 Å was observed in Supplementary Fig. 3, possibly due to the releasing of internal stresses from the tortured hard PBI chains in the double cross-linked 3D layered structures and formation of longer phosphate bridges (Supplementary Fig. 4).

The mechanism underlying membrane formation was elucidated through molecular dynamics (MD) simulations and radial distribution function (RDF) calculations, as depicted in Fig. 2m–o, Supplementary Fig. 5 and Supplementary Movie 1 and Supplementary Movie 2. Initially, DC-PBI polymer chains were uniformly distributed in PPA. However, the hydrolysis of PPA to PA triggered a sol-to-gel transition, resulting in noticeable phase separation due to the significant solubility difference of PBI in PPA and PA. Subsequent removal of PA revealed a 3D layered structure in the DC-PBI-G membrane. The oriented alignment of PBI nano-layers was validated by 2D SAXS images in Supplementary Fig. 6a, confirming the anisotropic nature of these 3D layered structures[21], likely due to rearranged forming stacks in benzimidazole rings parallel to the membrane surface[22,23]. The 3D model of DC-PBI-G structure with 3.9 Å and 9.6 Å inter-chain distances (Supplementary Fig. 7) was in good agreement with the WAXD analysis, suggesting larger intermolecular spaces and free volumes for PA storage. This unique architecture was anticipated to confer a combination of high acid doping, proton conductivity, and satisfactory gas barrier properties to the gel-state membranes.

Results in Table 2 revealed that gel-state PA-PBI PEMs with 3D layered structures had exceptionally high ADL$_T$ values (59.3-66.7 mol PA PRU$^{-1}$)[20], much higher than that of dense m-PBI-D (6.1 mol PA PRU$^{-1}$) made from conventional method with high polymer content. Among these, the double cross-linked DC-PBI-G with the largest $d_1$ spacing showed the highest ADL$_T$, ascribing to the incorporation of hydroxyl groups to anchor phosphate bridges and form a higher density of hydrogen bonding. Additionally, the introduction of a branched structure hindered the tight stacking of PBI polymer chains, thereby creating larger free volume for the retention of more PA.

## Effects of double cross-linked networks

The presence of HO- groups and the formation of phosphate bridges in the structures of both DC-PBI-G and HO-PBI-G could be supported by the emergence of two novel FT-IR peaks, appearing at 1540 cm$^{-1}$ attributed to the substituted benzene ring's vibration mode of the hydroxyl and carboxyl groups[15] and at 1357 cm$^{-1}$ corresponds to the bending vibration mode of hydroxyl groups as depicted in Fig. 3a, b, while the characteristic stretching vibration peaks of benzimidazole and imidazole appeared at 1620 cm$^{-1}$ and 1450 cm$^{-1}$, respectively[24,25]. Moreover, the appearance of the P-O-C peak at 1240 cm$^{-1}$ in DC-PBI-G and HO-PBI-G, in addition to those peaks observed for linear p-PBI-G, supported the formation of the phosphate bridge structures[15]. Such P-O-C peak at 1240 cm$^{-1}$ for DC-PBI-G membrane was intensified in DC-PBI-G-240 and weakened in DC-PBI-G-KOH, indicating phosphate bridges could be increased under high-temperature treatment and broken in KOH solution as revealed in Fig. 3b and Supplementary Fig. 8[26]. There was no change in the FT-IR spectra of p-PBI-G, p-PBI-G-240, and p-PBI-G-KOH (Fig. 3c), indicating no phosphate bridge exist in their linear PBI structures.

The anchoring of PA or PPA by the HO- groups and the formation of phosphate bridges in DC-PBI-G membranes could be further evidenced by EDS elemental mapping results shown in Fig. 3d–g, where phosphorus (P) and oxygen (O) elements were both uniformly distributed in their cross-sections, with significant increases observed in the contents of P (Fig. 3f) and O elements (Fig. 3g) in DC-PBI-G-240. Furthermore, X-ray photoelectron spectroscopy (XPS) analysis in Fig. 3h and Supplementary Table 3, gave a phosphorus-to-nitrogen (P/N) ratio of 0.23 for DC-PBI-G, which decreased to 0.06 in DC-PBI-G-KOH and significantly increased to 1.07 in DC-PBI-G-240. Conversely, p-PBI-G, p-PBI-G-240, and p-PBI-G-KOH showed a constant P/N ratio of around 0.03, corresponding to a minimal amount of PA anchored to the PBI framework (ADL$_A$) of around 0.05 mol PA PRU$^{-1}$.

The specific structure and composition of the constructed phosphate bridges were further elucidated using solid-state $^{31}$P nuclear magnetic resonance ($^{31}$P NMR) spectra, as depicted in Supplementary Fig. 4 and Supplementary Fig. 9. DC-PBI-G had predominately (100%) $P_1$ type phosphate bridges composed of one PA unit[27]. After 240 °C heat treatment, DC-PBI-G-240 showed longer types of phosphate bridges, consisting of $P_2$ (31%) and $P_3$ (46%), made of two or more than two PA units, with an average of 2.23 PA units per bridge. Such elongation of the phosphate bridges, associated with the expansion in $d_1$ spacing (10.2 Å, Supplementary Fig. 3), was probably caused by the releasing of internal stress from the tortured PBI chains bent in the double cross-linked networks. Based on the increase of P/N ratio from 0.23 to 1.07, it could be derived that the total amounts of phosphate bridges increased by 106% during these elongating processes. The anchoring of more PA in situ at 240 °C was primarily driven by the thermodynamic favoring of the reactions between PA molecules and HO-groups on the PBI chains.

The double cross-linking effect from the formation of phosphate bridges in HO-PBI-G membranes and further from the inclusion of TMA monomer in DC-PBI-G membranes could also be validated in the gel retention test after immersed in concentrated sulfuric acid (98%) for 10 days. As shown in Fig. 3i, linear m-PBI-D, p-PBI-G, and their respective heat-treated membranes were completely dissolved while HO-PBI-G and DC-PBI-G showed 73.2% and 77.1% of gel retention, indicating both phosphate bridge and branching of PBI chains had enhanced the membrane's cross-linking degree and also the resistance to the strong acid environment. After heat treatment, the gel retention rates of HO-PBI-G-240 and DC-PBI-G-240 were increased to 80.1% and 90.1%, respectively. Double cross-linking degrees could also be compared by the gel retention for the membranes shown in Fig. 3i from soaking in DMAc (99%) for 10 days, where DC-PBI-G and DC-PBI-G-240 showed the highest retention rates of 92.3% and 94.4%, respectively,

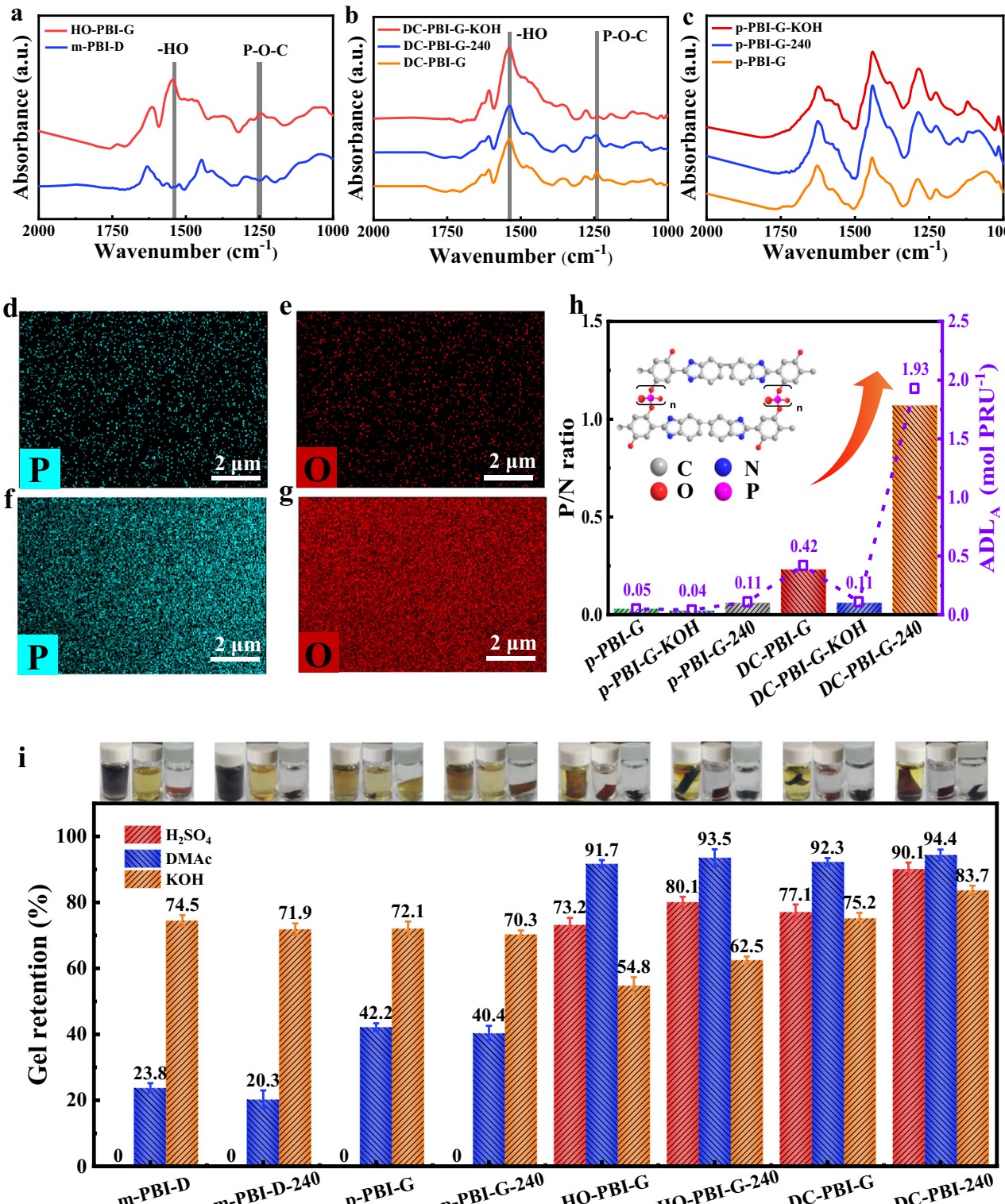

**Fig. 3 | Chemical structure, elemental characterization and physical property of PA-undoped membranes. a–c** FT-IR spectra of various membranes. **d, e** Elemental mapping of DC-PBI-G. **f, g** Elemental mapping of DC-PBI-G-240. **h** Comparison of surface P/N ratio and $ADL_A$ obtained from XPS. **i** Gel retention rate and photographs from 98% sulfuric acid, 1 M KOH and 99% DMAc at room temperature for 10 days. All error bars indicate the s.d. of at least three independent measurements.

while linear m-PBI-D, p-PBI-G, and their respective heat-treated membranes showed less than 50% of gel retention rates.

The membranes were soaked in KOH (1 M) for 10 days. The retention rate of m-PBI-D, p-PBI-G, and their heat-treated membranes were over 70%. However, the retention rate of HO-PBI-G was only 54.8%, attributed to the degradation of phosphate bridges in KOH[15]. On the other hand, the retention rate of DC-PBI-G with branched PBI chains reached 75.2% and could be further increased to 83.7% after heat treatment. DC-PBI-G-240 membranes had a high content of phosphate bridges but showed the least hydrolysis loss in KOH

treatment, indicating that not all the phosphate bridges were broken. The weight loss in KOH could be ascribed to the degradation of the imidazole rings in the PBI backbone, resulting in a weight loss of 25% to 30%, as shown in m-PBI-D and p-PBI-G membranes. An additional 10% weight loss came from the hydrolysis of the phosphate bridges, as observed in OH-PBI-G membranes. The 20% lower weight loss of DC-PBI-G and DC-PBI-G-240 membranes in KOH compared to HO-PBI-G and HO-PBI-G-240 could be assigned to their double cross-linked 3D layered structures that hindered the OH⁻ attacks on the backbones and phosphate bridges.

Collectively, the above results in FT-IR, EDS, XPS, and $^{31}$P NMR confirmed the formation of phosphate bridges in DC-PBI-G, which could be increased in numbers and elongated under 240 °C heating, and only partially hydrolyzed by KOH solution. DC-PBI-G membranes exhibit excellent resistance to acidic, alkaline, and organic solvent environments, as well as thermal stability (1% weight loss at 300 °C, see Supplementary Fig. 10). These results fully demonstrated the effects of the synergistic double cross-linked action from the phosphate bridge and rigid branched PBI chains.

## PA retention and creep resistance

PA loss significantly impacts PEMFC performance, particularly under harsh operational conditions such as high temperature or high humidity. To provide a more accurate PA retention evaluation, ADL, which combined both $ADL_T$ and $ADL_A$ (Supplementary Fig. 11), was considered in this study. The results in Fig. 4a, b highlighted a notable in-situ PA anchoring process in PA-doped DC-PBI-G. Consequently, its ability to anchor PA and establish extensive 3D hydrogen bonding networks contributed to a much higher ADL retention of 86.0%, still retaining 45.4 mol PA PRU⁻¹ even after 50 h at 240 °C (Fig. 4c, d),

compared to p-PBI-G's 18.8 mol PA PRU⁻¹ and 73.2%. At 80 °C/40% RH (Supplementary Fig. 12), DC-PBI-G also demonstrated greatly enhanced ADL retention of 77.7%, indicating its effective mitigation of PA loss.

To investigate the condensation behavior of PA molecules, we analyzed the liquid-state $^{31}$P NMR spectra of PA solutions with concentrations ranging from 85% to 115%. These spectra (Fig. 5a), consistent with existing literature[28], revealed three distinct peaks corresponding to orthophosphoric acid ($P_1$), pyrophosphoric acid ($P_2$), and triphosphoric acid or polyphosphoric acid ($P_3$)[27]. The solid-state $^{31}$P NMR of m-PBI-D-240 and p-PBI-G-240 showed more than 44% of PA had been dehydrated to form $P_2$ or $P_3$ after high-temperature treatment, while $^{31}$P NMR spectra of DC-PBI-G-240 membrane suggested that 96% of the PA molecules (Fig. 5b) had been safely retained without further dehydration or condensation (Supplementary Table 4)[27,28]. The observed effect can be attributed to the anchored PA via covalent bonds, resisting further reaction due to elevated activation energy. In-situ formed phosphonate bridges within 3D layer microstructure may also confine free PA motion and suppress their condensation. This interpretation was supported by the reduction in the proportion of $d_1$ spacing from 44% to 21% after high-temperature treatment (Supplementary Table 2).

High-temperature creep resistance of PEMs is crucial for practical applications in fuel cells (especially >200 °C). Over time, creep and stress relaxation could cause membrane thinning, resulting in out-leached PA, weakened contact between PEMs and the electrodes, leading to reduced proton conductivity, gas crossover, gradual performance decline and eventual failure of fuel cell[8]. As shown in Supplementary Table 5, m-PBI-D showed the lowest steady-state creep compliance $J^0_s$ (extrapolated to t = 0) at 180 °C and lowest ADL

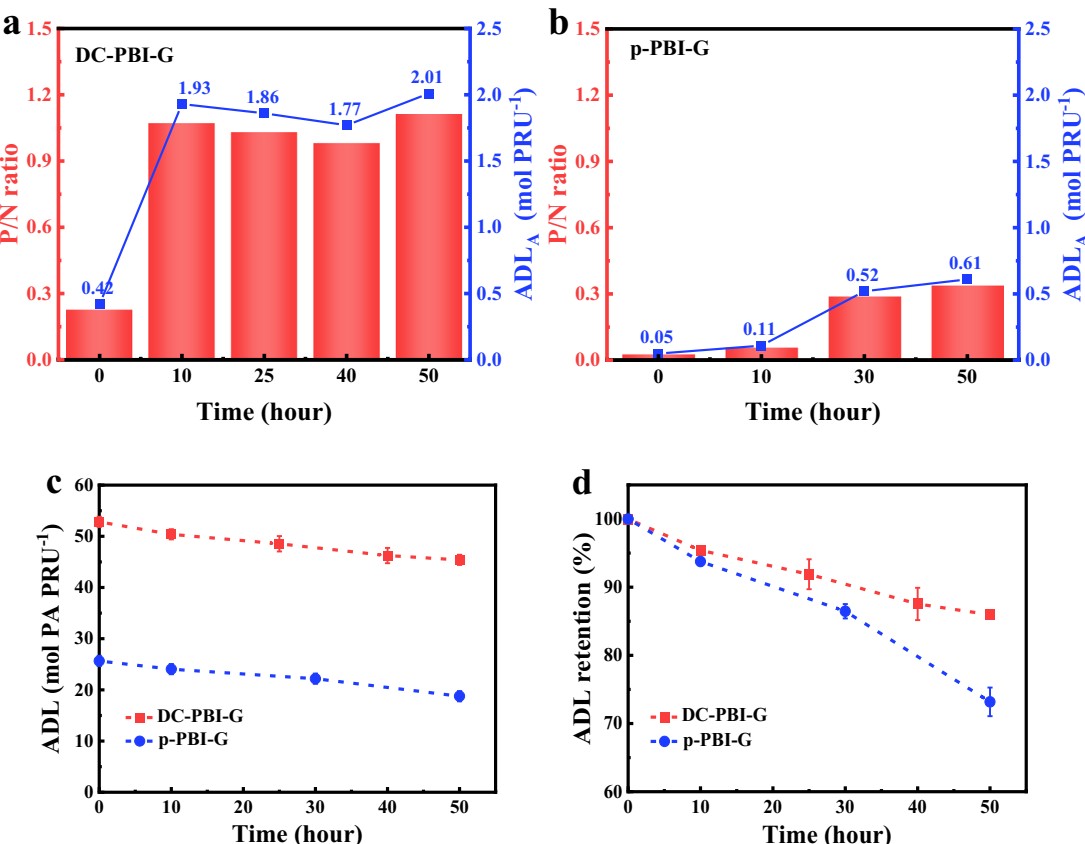

**Fig. 4 | In situ anchoring of PA and ADL retention of PA-doped membranes at 240 °C.** The changes in P/N ratio and $ADL_A$ over time for (**a**) DC-PBI-G and (**b**) p-PBI-G. **c** ADL values for both membranes. **d** Comparison of ADL retention between DC-PBI-G and p-PBI-G. All error bars indicate the s.d. of at least three independent measurements.

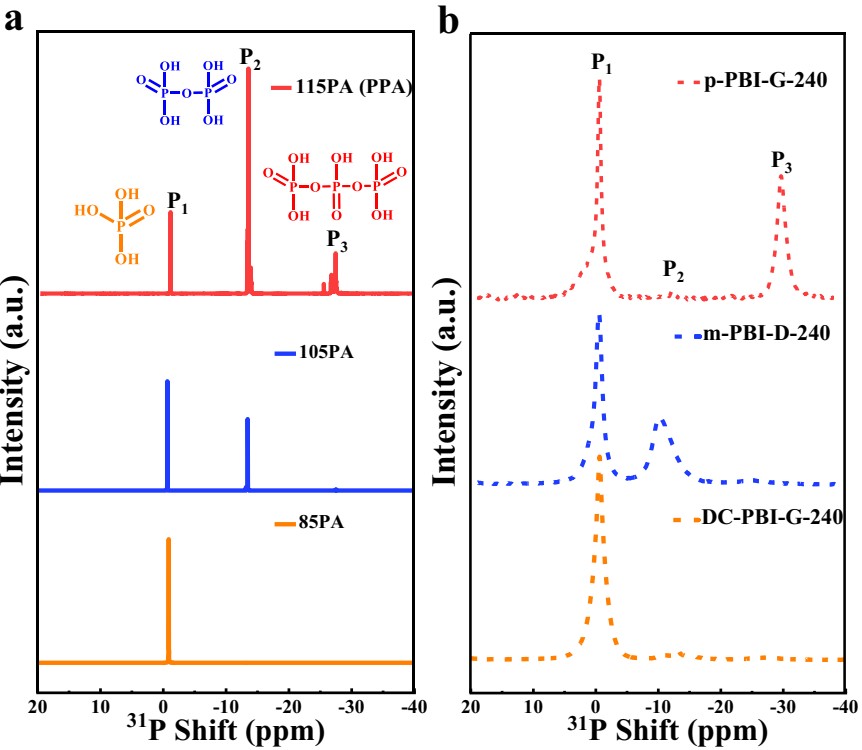

**Fig. 5 | $^{31}$P NMR spectra of PA solutions and doped membranes. a** Liquid-state $^{31}$P NMR spectra of 85% PA solution (85PA), 105% PA solution (105PA) and 115% PA solution (115PA or PPA), externally referenced to H$_3$PO$_4$. **b** Solid-state $^{31}$P NMR spectra of PA-doped DC-PBI-G-240, p-PBI-G-240 and m-PBI-D-240 membranes, externally referenced to (NH$_4$)$_2$HPO$_4$.

retention level at 240 °C, p-PBI-G had the highest J$^0$$_s$ with none crosslinked highly porous structure. From p-PBI-G to HO-PBI-G and DC-PBI-G, J$^0$$_s$ decreased greatly from 9.7 × 10$^{-6}$ Pa$^{-1}$ to 5.3 × 10$^{-6}$ Pa$^{-1}$ and 2.1 × 10$^{-6}$ Pa$^{-1}$ when ADL retention rates were steadily increased from 83.8 to 89.5 and 92.6% (Supplementary Table 5). Overall, on the basis of these results, the formation of a double cross-linked 3D layered structure not only enhanced acid retention levels but also mitigated PA dehydration and condensation, while preventing membrane creeping under high operating temperatures.

## Proton conductivity and stability

The through-plane proton conductivity of all PA-PBI membranes was evaluated at temperatures ranging from 20 °C to 240 °C under anhydrous conditions (Fig. 6a). p-PBI-G membrane presented much higher through-plane proton conductivity (0.278 S cm$^{-1}$ at 180 °C) than dense m-PBI-D membrane (6.1 PA PRU$^{-1}$ and 0.113 S cm$^{-1}$ at 180 °C) due to its ultra-high ADL (59.3 PA PRU$^{-1}$) and highly porous proton transport pathways (Fig. 7a). HO-PBI-G membrane exhibited higher through-plane proton conductivity than the p-PBI-G and m-PBI-D membranes when phosphate bridges provided extra channels for proton transportation (Fig. 7b). Incorporation of a rigid branched network further facilitated the formation of more pores and continuous three-dimensional hydrogen bonding networks, subsequently improving the proton conductivity performance of the DC-PBI-G membrane (Fig. 7c). The through-plane and in-plane proton conductivity of DC-PBI-G steadily increased and reached a value of 0.348 S cm$^{-1}$ and 0.395 S cm$^{-1}$ (Supplementary Fig. 13) at 220 °C, respectively. Furthermore, the proton conductivities of DC-PBI-G-240 surpassed those of p-PBI-G-240 (Supplementary Fig. 14).

The high-temperature proton conductivity stability (Fig. 6b) and retention (Fig. 6c) were compared. DC-PBI-G exhibited the best stability, retaining 72% of its initial conductivity after 100 h of testing at 240 °C, with a stabilized proton conductivity of approximately 0.241 S cm$^{-1}$, resulting from their remarkable resistance to creep and the ability to effectively anchor and retain a high level of PA. As shown in Fig. 6e, f, DC-PBI-G membranes developed in this work demonstrated superior proton conductivity compared to the most reported in the literature to date, having gel retention rates of over 90% in DMAc and stable performance above 200 °C. These could be further supported by the fact that the intrinsic proton conductivity of DC-PBI-G-240 excluded free PA under anhydrous conditions was two orders of magnitude higher than that of m-PBI-D-240 (Supplementary Fig. 15), indicating the potential novel proton transfer pathways (Fig. 7) over a large number of elongated phosphate bridges existing in the double cross-linked PBI networks.

## Fuel cell performance and durability

In Fig. 8a, b, DC-PBI-G based MEAs delivered an open circuit voltage exceeding 0.95 V and satisfactory gas barrier capability across all temperatures (Supplementary Fig. 16 and Supplementary Table 6), with a peak power density ascending to 1480 mW cm$^{-2}$ at 200 °C, and to impressive 1302 mW cm$^{-2}$ at 240 °C. Even with a reduced Pt loading from 1.0 mg cm$^{-2}$ to 0.5 mg cm$^{-2}$, the superior proton conductivity of DC-PBI-G ensured an excellent peak power density of 886 mW cm$^{-2}$ at 200 °C (Supplementary Fig. 17), as did DC-PBI-G-240 (Supplementary Fig. 18). These results resonated well with the high-frequency resistance trends depicted in Supplementary Fig. 19. Furthermore, DC-PBI-G exhibited remarkable fuel cell performance, achieving 636 mW cm$^{-2}$ when operated with methanol reformate gas (75% H$_2$, 24% CO$_2$ and 1% CO) and O$_2$ at 240 °C (Supplementary Fig. 20). DC-PBI-G exhibit noticeably higher power density than p-PBI-G and m-PBI-D membranes at 160 °C and 200 °C under both H$_2$/O$_2$ (Fig. 8c, d) and H$_2$/Air (Supplementary Fig. 21) conditions, confirming the efficacy of its dual-crosslinked 3D layered structures.

High-temperature durability tests were carried out under 220 °C and 0.2 A cm$^{-2}$ current load without external humidification, as shown

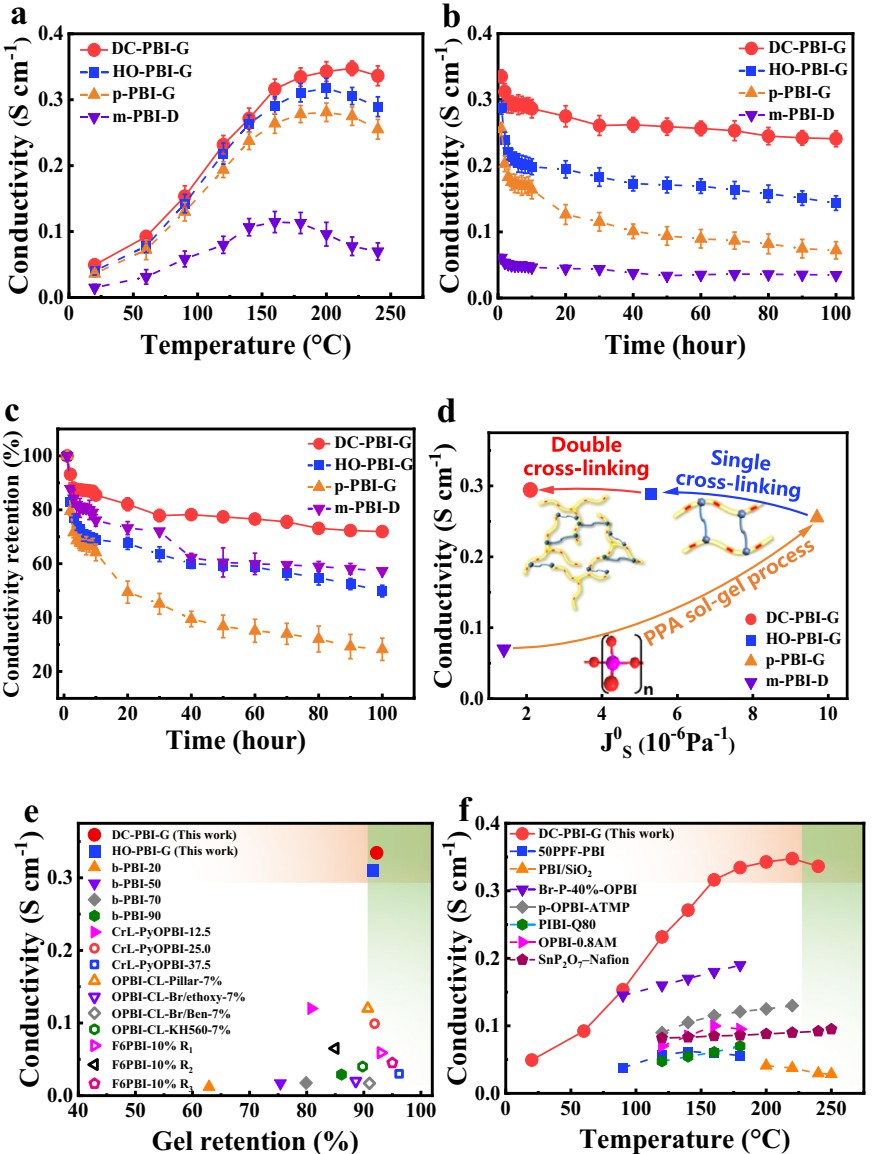

**Fig. 6 | Through-plane proton conductivity performance of PA-doped membranes. a** Proton conductivity at varied temperatures. **b** Proton conductivity stability at 240 °C. **c** Conductivity retention at 240 °C. **d** Optimization strategies diagram of this study. **e** Comparative analysis of proton conductivity versus gel content of the reported PBI membranes[14,35–37]. **f** Comparison of this work with reported PEMs in conductivity[38–45]. All error bars indicate the s.d. of at least three independent measurements.

in Fig. 8e. The p-PBI-G experienced notable voltage decay symptoms (0.53 mV h$^{-1}$) and severe high-temperature creep with noticeable cracks and holes in the MEA (Supplementary Fig. 22), leading to an operational failure in 150 h. In contrast, DC-PBI-G based fuel cell demonstrated a substantially lower voltage decay rate of merely 0.27 mV h$^{-1}$ over a 250-h testing period. Double cross-linking strategy enabled DC-PBI-G membranes to possess rigid 3D polymeric networks, with not only high pore volume for retaining large amount of PA, but also greatly improved stability under higher temperatures to avoid the out-leaching and dehydration of PA. Therefore, their FC performance above 200 °C was greatly improved.

At low temperatures (e.g., 40 °C), the DC-PBI-G based FC also displayed excellent output performance (443 mW cm$^{-2}$, in Supplementary Fig. 23), with a minimal decay rate of 8.97 μV h$^{-1}$ over 170 h. Following a more demanding accelerated stress test (AST) involving 100 start-up/shut-down cycles, it maintained a power density peak of 331 mW cm$^{-2}$, corresponding to a 75% retention (Supplementary

Fig. 24). DC-PBI-G membrane demonstrated superior fuel cell performance and operational flexibility under both low-temperature and high-temperature conditions to various representative PA-doped HT-PEMs (Fig. 8f and Supplementary Tables 7−9). Its peak power density surpassed that of the majority of previously reported systems, concordant with its high proton conductivity and structural stability.

## Discussion

This work delineated the regulation of proton conductivity and creep resistance of PEMs through the chemical modification of their structures, aiming at developing high-performance PA-doped PEMs applicable to high-temperature fuel cells. To this end, p-PBI-G membrane with open 3D layered structures was prepared by the PPA sol−gel method with improved ADL and proton conductivity but with compromised high-temperature creep resistance and performance stability. Double cross-linked PBI gel-state membranes (DC-PBI-G) with 3D layered structures were synthesized from the condensation

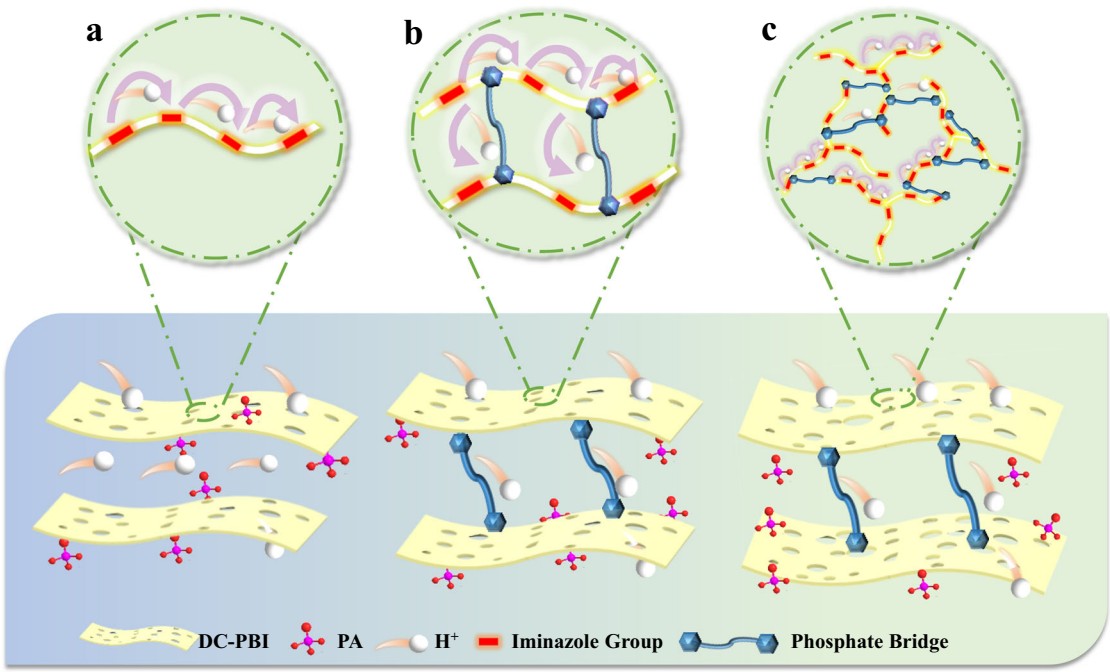

**Fig. 7 | Proton transfer mechanisms of PA-doped gel-state PBI membranes.** Proposed pathways for proton conduction in (**a**) p-PBI-G, (**b**) HO-PBI-G, and (**c**) DC-PBI-G.

polymerization of TAB, DHTA and TMA utilizing the PPA sol–gel method. HO- groups were built onto PBI networks to in-situ anchor PPA and PA molecules and yield extensive phosphate bridges while addition of tri-functional TMA monomer bestowed double cross-linking among the formed branched PBI polymeric chains, leading to the formation of highly stable DC-PBI-G 3D layered structures with improved ADL retention, increased proton conductivity and greatly reduced creeping above 200 °C.

In summary, DC-PBI-G demonstrated a remarkable acid doping level of 66.7 PA PRU$^{-1}$, excellent PA retention of 86% at 240 °C, impressive proton conductivity of 0.348 S cm$^{-1}$ at 220 °C, proton conductivity retention of 72% after 100 h at 240 °C, and low creep rate of 13.8 × 10$^{-12}$ Pa$^{-1}$ s$^{-1}$ and J$^0_s$ value of 2.1 × 10$^{-6}$ Pa$^{-1}$. The fuel cell utilizing DC-PBI-G achieved a peak power density of 1.48 W cm$^{-2}$ at 200 °C with a voltage decay rate of only 0.27 mV h$^{-1}$ at 220 °C.

## Methods
### Materials
3,3'-diaminobenzidine (TAB, ~97% purity) was purchased from Shanghai Macklin Biochemical Technology Co., Ltd. 2,5-dihydroxyterephthalic acid (DHTA, >97% purity) was purchased from Shanghai Bide Pharmaceutical Technology Co., Ltd. Trimesic acid (TMA, >97% purity) was purchased from Beijing Inokay Technology Co., Ltd. Terephthalic acid (TPA, >97% purity) was purchased from Shanghai Macklin Biochemical Technology Co., Ltd. Polyphosphoric acid (PPA, 115%) was purchased from Jinan Huifengda Chemical Co., Ltd. Phosphoric acid (PA, >85%) was purchased from China National Medicines Co., Ltd. The conventional PBI membrane (m-PBI-D) donated by PBI Performance company was imbibed in 85 wt% PA solution for further tests and comparison. The gas diffusion electrodes (GDEs) were supplied by BASF Fuel Cell, Inc.

### Polymer synthesis and membrane fabrication
In this study, three gel-state PBI-based membranes (that is, DC-PBI-G, HO-PBI-G, and p-PBI-G) were prepared via a PPA sol–gel approach, with detailed descriptions provided in the supplementary information. For the synthesis of DC-PBI-G, 3,3'-diaminobenzidine (TAB,1.0709 g), 2,5-dihydroxyterephthalic acid (DHTA, 0.9211 g), and trimesic acid (TMA,

0.0488 g) were dissolved in polyphosphoric acid (PPA, 120 g), then the reaction mixture was continuously stirred and purged with dry nitrogen in a 250 ml three-necked flask equipped with mechanical stirring. The reaction terminated at 190 °C for 20 h when a moderate viscosity was attained in the polymer solution. The resulting solution was then uniformly cast onto a glass substrate using a doctor blade, yielding a 235 μm thick gel-state PA-doped DC-PBI-G membrane through the sol–gel phase inversion at ambient temperature and 55%RH conditions for 24 h. The detailed fabrication process and chemical structures of the other three PA-doped gel-state PBI membranes were provided in the supplementary information and Supplementary Fig. 9.

### High-temperature treatment procedure
The PA-doped PBI membranes underwent curing in a 240 °C oven for 10 h and were gently wiped with tissue paper. These resulting membranes were denoted as "NAME-240," with "NAME" representing the specific membrane name.

### PA removal procedure
The PA-doped membranes were immersed in room-temperature deionized water. This process was repeated every 12 h until the pH of the deionized water bath reached neutrality. Subsequently, the resulting wet membranes were either air-dried or freeze-dried and referred to as PA-undoped PBI membranes.

### KOH treatment procedure
Following the removal of PA, the PA-undoped membranes were immersed in a 1 M KOH solution at room temperature for 10 h. They were then thoroughly rinsed in a series of deionized water baths to remove any excess KOH. Subsequently, the resulting wet membranes were dried and denoted as "NAME-KOH," with "NAME" representing the specific membrane name.

### Chemical structure and micromorphology
The chemical structure and elemental composition of the synthesized gel-state PA-undoped PBI membranes were analyzed using Fourier transform infrared spectrometry (FT-IR, iS50 Thermo Fisher Nicole, USA), X-ray photoelectron spectroscopy (XPS, Thermo ESCALAB

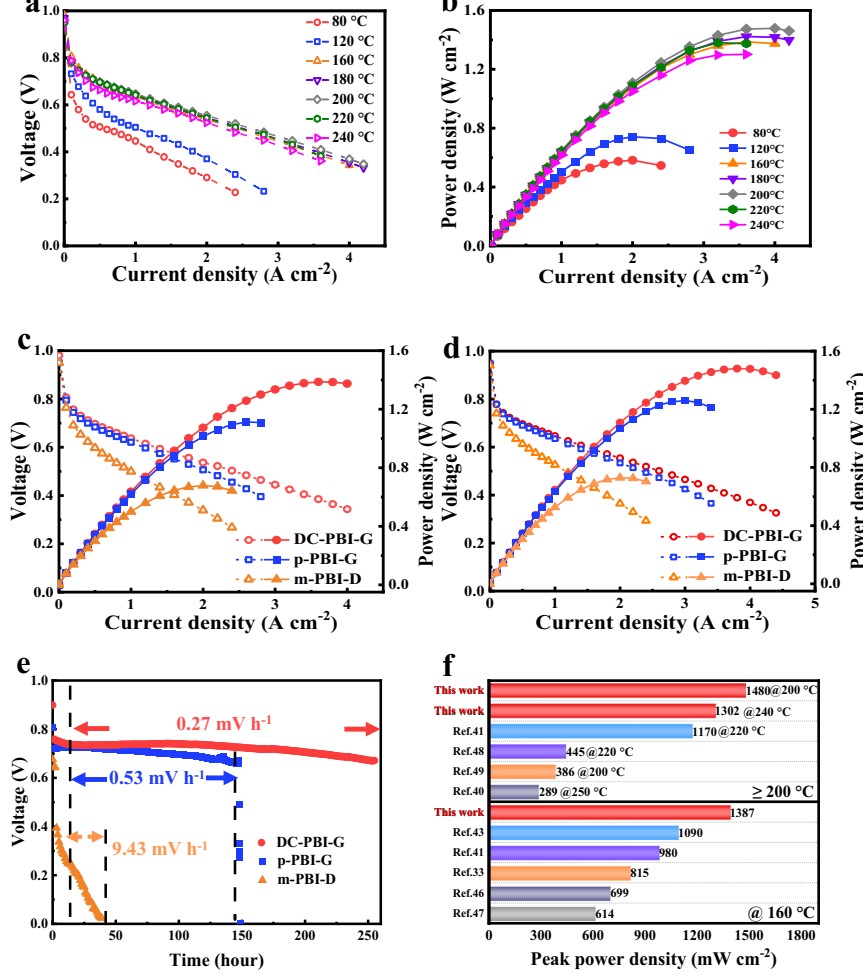

**Fig. 8 | Single-cell performance of PA-doped membranes under anhydrous conditions with dry $H_2$/$O_2$ at atmospheric pressure (a Pt loading of 1.0 mg cm$^{-2}$ on both sides). a** Polarization curves at different temperatures. **b** Power density curves at different temperatures. Single-cell performance comparison based on DC-PBI-G, p-PBI-G and m-PBI-D membranes at (**c**) 160 °C and (**d**) 200 °C. **e** Long-term durability test at 0.2 A cm$^{-2}$ and 220 °C. **f** Comparison of the peak power density of high-temperature (HT)-PEMFCs[33,40,41,43,46–49].

250XI, USA), and $^{31}$P nuclear magnetic resonance ($^{31}$P NMR, Bruker, Germany) spectroscopy. Micromorphology, elemental distribution, and d-spacing of polymer segments were evaluated via scanning electron microscopy (SEM, SU-8010, Hitachi, Japan), energy-dispersive X-ray spectrometry (EDX), wide-angle X-ray diffraction (WAXD, R-AXIS RAPID), and small angle X-ray scattering (SAXS). The specific surface area and pore volume of undoped PBI membranes were determined using the Brunauer-Emmett-Teller (BET) method (Micromeritics Instrument Corporation, USA). Further details are available in the supplementary information.

## Molecular simulation

Atomistic molecular dynamics (MD) simulations were conducted using the GROMACS[29] simulation package (version 2020.6) with the General Amber force field (GAFF2). Initially, 80 chains of the branched DC-PBI were randomly placed in a 12 × 12 × 12 nm$^3$ amorphous cubic cell. Subsequently, PA and PPA molecules were individually introduced into the cell. After undergoing thousands of energy minimization steps, the systems were equilibrated for 5 ns, followed by production runs of 50 ns under the NPT ensemble. Temperature coupling was maintained at 298 K using the Nose-Hoover method, while pressure coupling was sustained at 1 atm through the Parrinello-Rahman method. Non-bonded interactions employed a cutoff scheme of 1.2 nm, and long-range electrostatic interactions were managed using the Particle Mesh

Ewald method[30] with a Fourier spacing of 0.1 nm. Covalent bonds involving hydrogen atoms were constrained using the LINCS algorithm[31].

## Inherent viscosity (IV) and molecular weight ($M_w$) measurements

IV measurements were employed to estimate the relative molecular weights of PBIs due to their limited solubility. IVs were determined using a Cannon Ubbelohde viscometer within a temperature-controlled water bath set at 30 °C. The thoroughly neutralized and dried polymers were dissolved in concentrated sulfuric acid (98%). Prior to injection into the viscometer, solution samples were filtered through a 0.45 μm PTFE syringe filter. The calculation of $IV$ and $M_W$ was performed using the following equations:

$$IV = \ln\left(\frac{t}{t_0}\right)/C \qquad (1)$$

$$lg\left[M_W\right] = a\,lg\left[IV\right] + b \qquad (2)$$

where $C$ is the concentration of polymer solution (0.2 g dL$^{-1}$), $t$ is the time for the polymer solution to flow through the viscometer, $t_0$ is the time for concentrated sulfuric acid to flow through the viscometer, and parameter $a$ and $b$ are obtained from Huggins–Kraemer method

by Yuan et al.[32] ($a = 1.1369$, $b = 2.4084$). The reproducibility in measuring $IV$ was used to assess the relative error for this technique, and was found to be 1%.

## Membrane composition

PA-doped membranes, composed of PA, polymer, and water, underwent compositional analysis by acid-base titration method. After soaking in deionized water at 50 °C for 24 h, neutralization was achieved with a 0.1 M NaOH solution. Subsequently, the membranes were washed, dried, weighed, characterized by XPS. $ADL_T$, $ADL_A$, ADL and polymer content were calculated using the following equations:

$$Polymer \% = \frac{W_{dry}}{W_{PA-doped}} \cdot 100 \tag{3}$$

$$PA \% = \frac{M_{PA} \cdot V_{NaOH} \cdot C_{NaOH}}{2 \cdot W_{PA-doped}} \cdot 100 \tag{4}$$

$$ADL_T = \frac{V_{NaOH} \cdot C_{NaOH} \cdot M_{PBI}}{2 \cdot W_{dry}} \cdot 100 \tag{5}$$

$$ADL_A = \frac{M_N}{M_P} \cdot k \cdot \lambda \tag{6}$$

$$ADL = ADL_T + ADL_A \tag{7}$$

where $W_{PA-doped}$ is the initial weight of PA-doped membrane, $W_{dry}$ is the weight of the dry membrane after titration, $V_{NaOH}$ and $C_{NaOH}$ represent the volume and concentration of NaOH solution required for titration to the second equivalence point (pH = 9), $M_{PBI}$, $M_{PA}$, $M_N$ and $M_P$ denote the molar masses of the PBI repeat unit, PA, phosphorus element, and nitrogen element, respectively. The variable $k$ refers to the mass ratio of the phosphorus element and nitrogen element, while $\lambda$ denotes the number of nitrogen atoms in each repeating unit of PBI (with $\lambda = 4$).

## Gel retention rate

To evaluate cross-linking degree and resistance to various environments, we conducted a gel retention rate test on the PA-undoped membranes. Dry membranes were initially weighed ($W_0$) and then immersed in DMAc, KOH (1 M), and concentrated sulfuric acid (98%) at room temperature for 10 days. After immersion, the remaining membranes were thoroughly washed, dried, and reweighed ($W_t$). The gel retention rate was determined using the following equation:

$$Gel\ Retention\ Rate \% = \frac{W_t}{W_0} \cdot 100 \tag{8}$$

## ADL retention

Relevant PA-doped membranes underwent pre-treatment for 5 h at 120 °C to remove residual water. Subsequently, these samples were exposed to environments at either 240 °C/0% RH or 80 °C/40% RH. During the test, membranes were wiped to eliminate leached PA, and ADLs in both the initial and remaining membranes were quantified via acid-base titration and XPS measurement. The ADL retention rate of the membrane was evaluated using the following equation:

$$ADL\ Retention \% = \frac{ADL_t}{ADL_0} = \frac{ADL_{Tt} + ADL_{At}}{ADL_{T0} + ADL_{A0}} \cdot 100 \tag{9}$$

where $ADL_0$ and $ADL_t$ represent the initial and residual ADL of PA-doped membranes, respectively. $ADL_{T0}$ and $ADL_{Tt}$ are the initial and residual amounts of PA determined via acid-based titration method,

while $ADL_{A0}$ and $ADL_{At}$ denote the initial and residual amounts of anchored PA quantified by XPS, respectively.

## [31]P NMR analysis

To observe variations during the PA dehydration-condensation process, a series of PA solutions with concentrations of 85% (85PA), 105% (105PA) and 115% (115PA) were prepared, with 85PA and 105PA derived from the hydrolysis of PPA. The liquid-state [31]P NMR spectra were recorded at room temperature in DMSO-$d_6$ using a Bruker 400 MHz spectrometer equipped with a 5 mm PABBO BB/19F-1H/D probe. The zgpg30 pulse sequence was employed with a pulse width of 7.9 µs, a recycle delay of 2 s, and a total of 180 scans. Chemical shifts were referenced to $H_3PO_4$ (external standard) at 0 ppm.

To assess the PA status within membranes, PA-doped samples were subjected to a 10-h treatment at 240 °C followed by the removal of residual PA from the surface. To characterize the chemical structure and composition of phosphonate bridges, undoped DC-PBI-G and DC-PBI-G-240 samples were prepared by thoroughly removing doped PA molecules. Solid-state [31]P NMR spectra were recorded at room temperature using a Bruker Avance Neo 400WB spectrometer with a H13925_0025 (PH MAS DVT 400W1 BL4 N-P/F-H) probe, spinning at 8 kHz. The cross-polarization pulse sequence was employed with a pulse width of 3.13 µs, a recycle delay of 40 s, and a total of 180 scans. Chemical shifts were externally referenced to $(NH_4)_2HPO_4$ at 1.00 ppm.

## Proton conductivity

Proton conductivity of the membrane was assessed using a two-probe electrochemical impedance spectroscopy method with a LCR bridge potentiostat, spanning a frequency range of 100 Hz to 100 kHz with an amplitude of 5 mV[33,34]. A 6 cm × 2 cm rectangular membrane sample was assembled into a custom-designed impedance test cell (Supplementary Fig. 25). The cell was preheated to 180 °C at a ramp rate of 40 °C h⁻¹ to remove residual water within the membrane. Both through-plane and in-plane proton conductivity data reported in this study corresponded to the second heating cycle, ranging from 20 °C to 240 °C, which was evaluated based on the following equation:

$$\sigma = \frac{D}{R \cdot A} \tag{10}$$

where $R$ represents the ohmic resistance of the membrane determined from the Nyquist plot, $D$ is the distance between two platinum electrodes, $A$ is the membrane cross-sectional area in the case of the in-plane set-up, and the area of the electrodes in the case of the through-plane set-up.

## Fuel cell performance and durability

The membrane electrode assembly (MEA) consisted of a PA-doped PBI membrane sandwiched between two GDEs. The anode GDE featured Pt/C, while the cathode GDE utilized Pt alloy/C, both with a Pt loading of 1.0 mg cm⁻². Asymmetric MEAs were fabricated by hot pressing the membrane between two GDEs with Kapton frames at 150 °C and 20 MPa for 120 s. Fuel cell tests were conducted on a 3 × 3 cm² single cell using a commercial NEWARE electrochemical station. Prior to testing, a break-in period of at least 12 h involved operating the cell at 160 °C and 0.2 A cm⁻² with dry $H_2$ and $O_2$. Polarization curves were recorded from 25 °C to 240 °C with various fuel streams, including $H_2$ (>99.999%), premixed methanol reformate gas (75% $H_2$, 25% $CO_2$, and 1% CO), $O_2$ (>99.999%) and air. High-temperature and low-temperature long-term durability tests were conducted at 220 °C and 40 °C, respectively, with a current density of 0.2 A cm⁻². Additionally, constant temperature (40 °C) shut-down and start-up cycling accelerated stress tests (AST) were performed. Following an I-V test, the cell was powered off, and the fuel intake valves were closed. After a 4-min interval, the fuel intake valves were reopened, and the cell was allowed

to stabilize for another 4 min before initiating the next step I-V cycle. Hydrogen crossover measurements in single cell were conducted across a temperature range from room temperature to 220 °C. Potentiodynamic scans were performed at a rate of 4 mV s$^{-1}$, spanning from 0 to 0.8 V. Dry hydrogen and nitrogen gases, flowing at a rate of 400 sccm, were employed for the measurements under ambient pressure conditions. All the cell performance tests in this study were conducted under atmospheric conditions without humidification or backpressure. The electrochemical impedance spectra were recorded using an LCR bridge potentiostat. The minimum flow rate and stoichiometry of H$_2$ were 50 sccm and 1.2, respectively, while the minimum flow rate and stoichiometry of O$_2$ were 100 sccm and 2, respectively. The error range for fuel cell tests was approximately 2%.

## Data availability

Source data were provided with this paper. The authors declare that the data supporting the findings of this study were available within the paper and Supplementary information. Further data beyond the immediate results presented here were available from the corresponding authors on reasonable. Source data are provided with this paper.

## Code availability

This study did not generate any datasets.

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

## Acknowledgements

This work was supported by the National Natural Science Foundation of China (Grant No. NSFC-22209147) and Consulting and Research Program of the Chinese Academy of Engineering (Grant No. 2022-DZ-08). The authors gratefully acknowledge BASF New Business for the support of gas diffusion electrodes.

## Author contributions

L.Z. and F.H. designed the experiments. L.Z. conducted polymer synthesis, characterization, MEA fabrication, fuel cell testing, and data collection. M.J.L., D.Y.Z. and M.Y.T. helped with synthesis, characterization and data collection. T.Z.Z. performed the SEM measurements. C.J.G provided guidance for the work. F.H. and L.X.X. developed the concept and supervised the research. L.Z., F.H. and L.X.X. analyzed all experimental data and wrote the paper. All authors provided critical feedback, contributed to shaping the research and manuscript, and commented on the final version.

## Competing interests

The authors declare no competing interests.
