## [Peer Review File · Nature Communications]

Double Cross-linked 3D Layered PBI Proton Exchange Membranes for Stable Fuel Cell Performance Above 200°CREVIEWER COMMENTS

Reviewer #1 (Remarks to the Author):

The manuscript reported a DC-PBI-G membrane with cross-linked three-dimensional layered structure was prepared by combining cross-linking modification and sol-gel method.

However, this work lacks novelty. The modification method of introducing hydroxyl into PBI molecular chain to crosslink with PA molecule and the preparation of PA-PBI membrane by sol-gel method have been reported in relevant literatures. The DC-PBI-G membranes have good proton exchange membrane performance at 200-240 °C by combining these two methods, but it is not explained how to avoid the performance degradation caused by dehydration of PA at high temperature. The DC-PBI-G membranes lack more detailed and further discussion and characterization for comprehensive evaluation. Hence, I don't think it can be considered by Nature Communications.

Some specific comments are listed as follows:

1. Page 3, Line 78-83: (i) "HO- groups were built onto PBI networks to anchor PPA and PA molecules and ...". In the results and discussion section and Supplementary Fig. 7., it is shown that PA is crosslinked with OH- without mentioning PPA. Therefore, the crosslinking structure of the membrane should be further verified and the specific composition of PA and PPA in the constructed phosphoric acid bridge should be characterized. (ii) "... while addition of ti-functional TMA monomer bestowed double cross-linking among the formed branching PBI polymeric chains...". The addition of TMA in PBI synthesis can only construct branched PBI polymers to form more crosslinked structures, without adding other types of functional groups for crosslinking. The concept of double crosslinking mentioned in the article needs to be reconsidered. In addition, the crosslinking degree of each crosslinked membrane should be determined.
2. Page 3, Scheme. 1: (i) The molecular structure and molecular weight of m-PBI and p-PBI are different, so it is difficult to accurately evaluate the advantages of the layered membrane structure prepared by the sol-gel method. (ii) Scheme 1b does not clearly indicate the difference between DC-PBI-G and DC-PBI-G-240. In addition, why is PA anchored in situ at 240 °C, and how to avoid dehydration polymerization of PA at this temperature? The amount of PA anchored during this process should be measured.
3. Page 6, Line 135-139: "...as shown in Fig. 1f-h with much higher porosity, pore volume, and surface area (Supplementary Table 1)". Considering that the prepared membrane has high porosity, it is necessary to measure the permeability of gas or fuel through the membrane.
4. Page 7, Fig. 1: The WAXD characterization of DC-PBI-G-240 membrane is missing to analyze the change of polymer chain spacing after PA anchoring. In addition, DC-PBI-G-240 membrane also lacks characterization data on phosphoric acid doping, proton conductivity and fuel cell performance. Therefore, the logic of article arrangement needs to be reconsidered.
5. Page 8, Table 2: Methods and parameters for calculating molecular weight using intrinsic viscosity should be provided.
6. Page 10, Line 230-235: (i) The P/N ratio obtained from XPS and FT-IR spectra proved that the PA bridge in the membrane was increased after branched chain modification and 240 °C heat treatment, but the hydrolysis loss of PA bridge in DC-PBI-G-240 membrane after soaking in KOH was the least. This opposite phenomenon needs to be explained carefully. (ii) Supplementary Figure 3: the TGA curves of all membranes should be supplemented, and the process and causes of thermal weight loss should be described in detail. It can be seen from the TGA curve that the membrane starts to lose weight at about 210 °C, so the thermal stability of the membrane is difficult to ensure stable operation at

240 °C.

7. Page 11, Table 3: High humidity during fuel cell operation is the main reason for the loss of PA from PEM. Therefore, the relative humidity should be considered in the process of ADL retention test.

8. Page 13, Fig. 4: As the proton conductivity test results of m-PBI-D, the operating temperature of PA-PBI membrane is limited below 200 °C due to the dehydration polymerization of PA. However, the proton conduction of p-PBI-G membrane prepared by sol-gel method can be realized above 200 °C, and the dehydration polymerization amount of PA in the membrane should be investigated above 200 °C. The reason for the better proton conduction stability of DC-PBI-G membrane at 240 °C should be elaborated in detail. The data in Fig. 4b and 4c are duplicated, and one of them is retained.

Reviewer #2 (Remarks to the Author):

The manuscript by Xue and coworkers reported a novel double cross-linked polybenzimidazole (PBI) membranes for high temperature proton exchange membrane fuel cells (HT-PEMFCs) to operate above 200 °C. The first cross-link comes from tri-functional monomers, and the second cross-link comes from phenol-based phosphoric acid condensation. The two types of interactions were clearly elucidated by comparing with the corresponding control samples. By combining with the sol-gel membrane forming techniques, the resultant PBI membranes show one of the highest ionic conductivity and HT-PEMFC peak power density among literatures. More importantly, the authors show stable HT-PEMFC performance at 220 °C. I think this manuscript is suitable to be published in Nature Communications after making some minor revisions.

1. On lines 160-162, the authors claimed that “Additional cross-linking caused by the branching PBI chains had further expanded d1 peak spacing to 9.5 Å in DC-PBI-G PEMs, resulting in larger free pore volumes for higher acid doping levels (ADL).” It seems the authors correlated the physical meaning of WAXD d1 and d2 spacing with the free pore volume, which is a little confusing. From Fig. 1 f-h, the inter-layer spacing should be much larger than the ~9.5 Å length scale. May the authors explain this in more detail?

2. The stacked FT-IR spectra in Fig. 2b-c and the explanation of the potential P-O-C peak at 1240 cm⁻¹ is also confusing. From lines 187-192, the authors claimed “Such P-O-C peak at 1240 cm⁻¹ for DC-PBI-G membrane was intensified in DC-PBI-G-240 and weakened in DC-PBI-G-KOH (Fig. 2b)” and “There was no change in the FT-IR spectra of p-PBI-G, p-PBI-G-240 and p-PBI-G-KOH (Fig. 2c)”. However, I see that the 1240 cm⁻¹ peak intensity change in Fig. 2b is nearly negligible, while there are obvious changes from 1000-1250 cm⁻¹ in Fig. 2c. Please check and explain this point.

3. The authors should clarify the “±” numbers in all the Tables, explaining they are the average of how many samples and whether they are standard deviation numbers.

4. The language presentation needs to be polished for this manuscript before publication. For example, there are multiple places showing “ti-functional TMA”, while it should be “tri-functional TMA”; line 55, it should be “lose performance” instead of “lost performance”. In addition, the authors should unify the use of hyphens, such as “cross-link” or “crosslink”, “FT-IR” or “FTIR”.

Reviewer #3 (Remarks to the Author):

This manuscript reported the PA-doped gel-state PBI membranes with double cross-linked three-dimensional layered structures, prepared via the PPA sol-gel process. The optimized membrane demonstrates excellent proton conductivity, along with successful application in high-temperature fuel cells above 200°C, yielding impressive power density results. While this work is interesting and suitable for publication in Nature Communications, there are certain aspects that require clarification and modification before acceptance.

1. In reference to the paragraph on lines 134-141, please provide a more detailed explanation of the mechanism behind the formation of the layered structure and its impact on membrane performance. Additionally, elucidate the rationale for investigating this particular aspect of performance.
2. On lines 18-19, it remains unclear whether the primary objective of the strategies employed in this study is to mitigate PA dehydration and condensation under high-temperature conditions or to prevent PA leaching. We believe further clarification is needed.
3. If the aim is to mitigate PA condensation at high temperatures, consider incorporating phosphorus NMR spectra for validation. If the emphasis is on preventing PA leaching, please include additional data on low-temperature FC performance and the shut-down/start-up cycling operations.
4. Please elucidate the influence of high proton conductivity on device performance and the potential reduction in Pt loading. If the study covers not only high-temperature aspects, we recommend evaluating low-temperature durability as well. Additionally, the current 100-hour durability test in Fig.5e is inadequate to meet contemporary HT-PEMFC requirements.
5. Regarding the layered microstructure of the membranes, please provide a comparison of proton conductivity along both the in-plane and through-plane directions.
6. The language in the abstract needs to be polished to enhance the emphasis on the key findings and core contributions of the work.

Manuscript ID: NCOMMS-23-46954A

Title: " PBI-based Proton Exchange Membranes with Double Cross-linked 3D Layered Structures for Stable Fuel Cell Performance Above 200 °C "

REVIEWER COMMENTS

Reviewer #1 (Remarks to the Author):

The manuscript reported a DC-PBI-G membrane with crosslinked three-dimensional layered structure was prepared by combining cross-linking modification and sol-gel method. However, this work lacks novelty. The modification method of introducing hydroxyl into PBI molecular chain to crosslink with PA molecule and the preparation of PA-PBI membrane by sol-gel method have been reported in relevant literatures. The DC-PBI-G membranes have good proton exchange membrane performance at 200-240 °C by combining these two methods, but it is not explained how to avoid the performance degradation caused by dehydration of PA at high temperature. The DC-PBI-G membranes lack more detailed and further discussion and characterization for comprehensive evaluation. Hence, I don't think it can be considered by Nature Communications. Some specific comments are listed as follows:

1. Page 3, Line 78-83: (i) "HO- groups were built onto PBI networks to anchor PPA and PA molecules and ...". In the results and discussion section and Supplementary Fig. 7., it is shown that PA is crosslinked with OH- without mentioning PPA. Therefore, the cross-linking structure of the membrane should be further verified and the specific composition of PA and PPA in the constructed phosphoric acid bridge should be characterized. (ii) "... while addition of tri-functional TMA monomer bestowed double cross-linking among the formed branching PBI polymeric chains...". The addition of TMA in PBI synthesis can only construct branched PBI polymers to form more

crosslinked structures, without adding other types of functional groups for cross-linking. The concept of double cross-linking mentioned in the article needs to be reconsidered. In addition, the crosslinking degree of each cross-linked membrane should be determined.

Response: Thank you for raising these issues.

(i) To provide further clarity, we conducted additional characterization of the specific structure and composition of the constructed phosphate bridges using solid-state ^{31}P NMR spectra, as illustrated in Supplementary Fig. 4 and Supplementary Fig. 9. DC-PBI-G had predominately (100%) P_1 type phosphate bridges composed of one PA unit¹. After 240 °C heat treatment, DC-PBI-G-240 showed longer types of phosphate bridges, consisting of P_2 (31%) and P_3 (46%), made of two or more than two PA units, with an average of 2.23 PA units per bridge. Such elongation of the phosphate bridges, associating with the expansion in d_1 spacing (10.2 Å, Supplementary Fig. 3), was probably caused by the releasing of internal stress from the tortured PBI chains bent in the double cross-linked networks.

Supplementary Figure 4. Solid-state ^{31}P NMR spectra of PA-undoped (a) DC-PBI-G and (b) DC-PBI-240, externally referenced to $(\text{NH}_4)_2\text{HPO}_4$ (solid).

Supplementary Figure 9. (a) Synthesis and (b) photographs of PA-doped DC-PBI-G, HO-PBI-G, and p-PBI-G membranes. The manufacturing of membranes was scaled up to about 10 cm × 35 cm.

(ii) Thank you for providing this valuable suggestion. The addition of TMA in PBI synthesis can not only construct branched PBI polymers, but also bring in the possibility for the PBI to be cross-linked each other's for having multiple branching points. Phosphate bridges within DC-PBI-G further facilitates the second cross-linking mechanism for the polymeric networks as revealed in Supplementary Fig. 9 and Scheme I.

In our study, the term "double cross-linking" refers to the presence of two distinct types of cross-links in the gel-state DC-PBI-G PEM. Firstly, there are covalent cross-linking phosphate bridges formed by the HO- groups on PBI and PA or PPA molecules. Secondly, there is a branched cross-linking structure created by introducing branched TMA monomers during PBI synthesis. These branches contribute to additional cross-linking within the PBI backbones, resulting in a 3D framework of PBI. Therefore, to distinguish between these two different types of cross-linking and emphasize the synergistic effect of the branched structure and phosphate bridges as key factors in membrane properties and fuel cell performance, we would like to retain the term "double cross-linking" in our work.

The cross-linking degree of the polymer membranes could be assessed by their gel retention weight percentage in solvents. Solvents such as sulfuric acid (H_2SO_4) or dimethylacetamide (DMAc) are often used for this purpose. As shown in Fig. 2i, non-crosslinked m-PBI-D and p-PBI-G dissolved completely in H_2SO_4 , while the single-crosslinked HO-PBI-G and double-crosslinked DC-PBI-G retained 73.2% and 77.1% of their initial weight, respectively. Similarly, in DMAc, the gel retention fraction for m-PBI-D and p-PBI-G was only 23.8% and 42.2%, while HO-PBI-G and DC-PBI-G retained 91.7% and 92.3% of their initial weight, respectively. These results indicated the successful establishment of cross-linked structures within the membranes, and the introduction of branched structures had further enhanced the degree of cross-linking.

Fig. 2 | Chemical structure, elemental characterization and physical property of PA-undoped membranes. (i) Gel retention rate from 98% sulfuric acid, 1 M KOH and 99% DMAc at room temperature for 10 days.

2. Page 3, Scheme. I: (i) The molecular structure and molecular weight of m-PBI and p-PBI are different, so it is difficult to accurately evaluate the advantages of the layered membrane structure prepared by the sol-gel method. (ii) Scheme 1b does not clearly indicate the difference between DC-PBI-G and DC-PBI-G-240. In addition, why is PA anchored in situ at 240 °C, and how to avoid dehydration polymerization of PA at this temperature? The amount of PA anchored during this process should be measured.

Response: Thank you for raising these issues.

(i) We agree with the reviewer that the molecular structure and molecular weight of m-PBI and p-PBI are different, so it is difficult to accurately evaluate the advantages of the layered membrane structure prepared by the sol-gel method. However, it's worth noting that producing p-PBI membranes through conventional methods is unfeasible because of their poor solubility in organic solvents. Consequently, commercial m-PBI membranes prepared conventionally have been commonly selected as a standard for comparison in the literature. Furthermore, previous studies^{2,3} have extensively compared dense m-PBI and gel-state m-PBI to assess and validate the advantages of

the gel-state microstructure. These comparisons have evaluated various aspects such as acid doping level, proton conductivity, and cell performance, with the corresponding results presented in the following figures.

Polybenzimidazole Fuel Cell Technology: Theory, Performance, and Applications, Table 1 Comparison of conventionally imbibed *m*-PBI vs. *m*-PBI synthesized from the PPA process [16]

IV ^a (dl g ⁻¹)	Film process	Polymer (wt%)	PA (wt %)	Water (wt%)	PA/PBI (molar ratio)	Proton diffusion coefficient ^b (cm ² s ⁻¹)	Conductivity ^c (S cm ⁻¹)
0.89	Conventionally imbibed	15.6	60.7	23.7	12.2	10 ⁻⁷	0.048
1.49	PPA process	14.4	63.3	22.3	13.8	3 × 10 ⁻⁶	0.13

^a Inherent viscosity (IV) was measured at a polymer concentration of 0.2 g dl⁻¹ in concentrated sulfuric acid (96%) at 30 °C, using a Canon Ubbelohde viscometer

^b Estimation of upper bound for conventionally imbibed *m*-PBI at 180 °C; PPA-prepared *m*-PBI measured at 180 °C

^c Measured at 160 °C after an initial heating to 160 °C to remove water

FIGURE 4 Ionic conductivities for heat-treated PA-doped *m*-PBI membranes at different acid doping levels for conventionally imbibed membranes [3 ± 0.3 (-Δ-), 4 ± 0.3 (-□-), and 5 ± 0.3 PA/PRU (-○-)] and sol-gel membranes [3 ± 0.2 (-▲-), 5 ± 0.2 (-■-), 6 ± 0.3 (-◆-), and 8 ± 0.7 PA/PRU (-▶-)].

Table 1. PBI Membrane Composition

membrane	monomer charge (wt %)	IV (dL/g)	polymer content (wt %)	PA content (wt %)	PA/PBI r.u. (molar ratio)
para -PBI (PPA)	2	3.3 ± 0.03	5.6 ± 0.08	57.3 ± 0.2	32.3 ± 0.06
m/p -PBI (PPA)	10	1.8 ± 0.02	16.8 ± 0.17	57.6 ± 0.6	10.8 ± 0.15
meta -PBI (PPA)	9	1.5 ± 0.02	18.5 ± 0.08	55.2 ± 0.2	9.5 ± 0.06
meta -PBI (conv)		1.0 ± 0.01	27.7 ± 0.35	53.1 ± 0.3	6.04 ± 0.12

Figure 3. Proton conductivities of the *para*-PBI, *m/p*-PBI copolymer, and *meta*-PBI membranes prepared by the PPA process. *Meta*-PBI prepared via the conventional imbibing process is included for reference (a relative error of ~5%).

Figure 5. Electrochemical hydrogen separation polarization curves using humidified pure hydrogen with a stoichiometry of 1.25 and a constant RH of 1.6% at 160 °C (a relative error of ~2%).

(ii) PA-doped DC-PBI-G-240 was derived from the heat treatment of PA-doped DC-PBI-G at 240 °C for 10 hours. Such high-temperature treatment enabled the reaction of HO- groups on PBI with the doped PA molecules and PPA chains, to form phosphonate bridges. As a result, the major differences between DC-PBI-G and DC-PBI-G-240 are the amount of PA anchored in the phosphonate bridges. X-ray photoelectron spectroscopy (XPS) in Fig. 2h revealed the increase of phosphorus-to-nitrogen (P/N) ratio from 0.23 to 1.07 for DC-PBI-G-240, corresponding to an increase of ADL_A from 0.42 mol to 1.93 mol PA/PRU. Thus, the total amounts of phosphate bridges had also been increased by 106% during these elongating processes. Furthermore, solid-state ³¹P NMR spectra in Supplementary Fig. 4 indicated the part of the phosphate bridges (P₁) in DC-PBI-G had been elongated to pyrophosphoric acid

(31%) and triphosphoric acid or PPA (46%) in DC-PBI-G-240 after the heat treatment. The anchoring of more PA in situ at 240 °C was mainly driven the thermodynamic favoring of reactions between PA molecules and the HO- groups on the PBI chains. Scheme 1b and Supplementary Fig. 9 have been updated to provide clearer clarification.

Fig. 2 | Chemical structure, elemental characterization and physical property of PA-undoped membranes. (d)-(e) Elemental mapping of DC-PBI-G. (f)-(g) Elemental mapping of DC-PBI-G-240. (h) Comparison of surface P/N ratio obtained from XPS.

Supplementary Figure 4. Solid-state ³¹P NMR spectra of PA-undoped (a) DC-PBI-G and (b) DC-PBI-240, externally referenced to (NH₄)₂HPO₄ (solid).

Scheme. I | Chemical structures and membrane preparation process. (a) Chemical structures of PBI membranes involved in this study. (b) Preparation process of PA-doped gel-state DC-PBI-G and DC-PBI-G-240 membranes.

Supplementary Figure 9. (a) Synthesis and (b) photographs of PA-doped DC-PBI-G, HO-PBI-G, and p-PBI-G membranes. The manufacturing of membranes was scaled up to about 10 cm × 35 cm.

Furthermore, the dehydration and condensation among PA molecules can be avoided by retaining them inside the stable double cross-linked structures. To investigate the condensation behavior of PA molecules, we analyzed the liquid-state ^{31}P NMR spectra of PA solutions with concentrations ranging from 85% to 115%. These spectra (Fig. 4a), consistent with existing literature⁴, revealed three distinct peaks attributed to orthophosphoric acid (P_1), pyrophosphoric acid (P_2), and triphosphoric acid or polyphosphoric acid (P_3)¹. The solid-state ^{31}P NMR of m-PBI-D-240 and p-PBI-G-240 showed more than 44% of PA had been dehydrated to form P_2 or P_3 after high temperature treatment, while ^{31}P NMR spectra of DC-PBI-G-240 membrane suggested that 96% of the PA molecules (Fig. 4b) had been safely retained without further dehydration or condensation (Supplementary Table 4)^{1,4}. The observed effect can be attributed to anchored PA via covalent bonds, resisting further dehydration reaction due to elevated activation energy. In-situ formed phosphonate bridges within the 3D layer microstructure could also confine free PA motion and suppress their condensation. These results indicated that PA molecules could be preserved better in the double cross-linked structures of DC-PBI-G.

Fig. 4 | ^{31}P NMR spectra. (a) Liquid-state ^{31}P NMR spectra of 85% PA solution (85PA), 105% PA solution (105PA) and 115% PA solution (115PA or PPA), externally referenced to H_3PO_4 . (b) Solid-state ^{31}P NMR spectra of PA-doped DC-PBI-G-240, p-PBI-G-240 and m-PBI-D-240 membranes, externally referenced to $(\text{NH}_4)_2\text{HPO}_4$.

Supplementary Table 4. The relative integrated peak intensities of ^{31}P NMR spectra.

Samples	P ₁	P ₂	P ₃
115PA (PPA)	5	63	32
105PA	53	45	2
85PA	100	0	0
PA-doped p-PBI-G-240	56	2	42
PA-doped m-PBI-D-240	56	42	2

3. Page 6, Line 135-139: "...as shown in Fig. 1f-h with much higher porosity, pore volume, and surface area (Supplementary Table 1)". Considering that the prepared membrane has high porosity, it is necessary to measure the permeability of gas or fuel through the membrane.

Response: Thank you for raising the issue. The H₂ crossover current density and diffusion rates were measured and presented in Supplementary Fig. 16 and Supplementary Table 6. The hydrogen crossover of DC-PBI-G remained much lower than those of Nafion[®] 211⁵ and reported PBI data⁶, although slightly higher than those of dense m-PBI-D and p-PBI-G membranes. This satisfactory gas barrier properties of the PA-doped DC-PBI-G membrane can be attributed to its unique 3D layered structure and anisotropic nature.

Supplementary Figure 16. H₂ crossover of PA-doped PBI membranes. (a) Equivalent H₂ crossover current densities of the fuel cell based on PA-PBI membrane as a function of operating temperature with ambient pressure at 0.5 V. (b) Linear voltammetry sweeps for Nafion[®] 211⁵ at 80 °C, PBI⁶ at 160 °C, and DC-PBI-G at both 80 °C and 160 °C.

Supplementary Table 6. H₂ crossover current densities and crossover rates of the fuel cell based on PA of operating temperature with ambient pressure at 0.5V.

Equivalent H ₂ crossover current at indicated temperatures at 0.5V						
	6 °C	40 °C	80 °C	160 °C	220 °C	Crossover rates
DC-PBI-G	0.76mA cm ⁻²	0.74mA cm ⁻²	1.05mA cm ⁻²	2.05mA cm ⁻²	3.32mA cm ⁻²	12.2μA cm ⁻² °C ⁻¹
p-PBI-G	0.19mA cm ⁻²	0.55mA cm ⁻²	0.34mA cm ⁻²	1.18mA cm ⁻²	3.99mA cm ⁻²	15.7μA cm ⁻² °C ⁻¹
m-PBI-D	0.30mA cm ⁻²	0.35mA cm ⁻²	0.56mA cm ⁻²	0.81mA cm ⁻²	0.94mA cm ⁻²	3.20μA cm ⁻² °C ⁻¹

4. Page 7, Fig. 1: The WAXD characterization of DC-PBI-G-240 membrane is missing to analyze the change of polymer chain spacing after PA anchoring. In addition, DC-PBI-G-240 membrane also lacks characterization data on phosphoric acid doping, proton conductivity and fuel cell performance. Therefore, the logic of article arrangement needs to be reconsidered.

Response: Thank you for your comments. WAXD spectra analysis on the DC-PBI-G-240 membrane have been incorporated into the manuscript. The cross-linking induced by phosphate bridges has led to a reduction in the d₂ packing spacing from 3.9 Å in p-PBI-G to 3.5 Å in HO-PBI-G. Furthermore, it has expanded the d₁ spacing from 8.4 Å in p-PBI-G to 8.9 Å in HO-PBI-G. The additional cross-linking caused by the branched PBI chains has further expanded the d₁ peak spacing to 9.5 Å in DC-PBI-G PEMs. After subjecting the membrane to high-temperature treatment at 240 °C, there was a further enlargement in the d₁ spacing (10.2 Å, Supplementary Fig. 3) for DC-PBI-G-240, possibly due to the releasing of internal stresses from the tortured hard PBI chains in the double cross-linked 3D layered structures and formation of longer phosphate bridges (Supplementary Fig. 4).

Supplementary Figure 3. WAXD spectra of undoped DC-PBI-G-240.

Supplementary Table 2. Comparison of WAXD peak position, area ratio and d-spacing.

	m-PBI-D	p-PBI-G	HO-PBI-G	DC-PBI-G	DC-PBI-G-240
Peak 1 2θ ($^\circ$)	/	10.53	9.94	9.32	8.77
Peak 1 Area ratio (%)	/	37.6	60.0	44.1	21.4
Peak 1 d-spacing (\AA)	/	8.4	8.9	9.5	10.2
Peak 2 2θ ($^\circ$)	23.01	22.48	25.17	24.46	24.7
Peak 2 Area ratio (%)	100.0	62.4	40.0	55.9	78.6
Peak 2 d-spacing (\AA)	3.8	3.9	3.5	3.6	3.6

Furthermore, PA-doped DC-PBI-G-240 was comprehensively characterized, including membrane composition, proton conductivity, and fuel cell performance. The results are summarized in Table 2 and Supplementary Fig. 14 and 24. Its polymer content and PA concentration increased, while the ADL_T value slightly decreased after high-temperature treatment. Its proton conductivity was much higher than that of p-PBI-G-240, maintaining excellent and stable FC performance above 200°C .

Table 2 | Polymer characteristics and PA-doped PBI membrane composition.

PA-doped membrane	Polymer content (wt%)	PA content (wt%)	ADL_T^a (PA/PRU)	IV (dL/g)	M_w^b (kDa)
m-PBI-D	27.6 ± 2.3	56.9 ± 5	6.1 ± 1	1.8	93.8
p-PBI-G	3.2 ± 0.5	58.7 ± 6	59.3 ± 2	3.1	174.1
HO-PBI-G	3.2 ± 0.4	60.2 ± 6	63.0 ± 2	N/A ^c	N/A ^c
DC-PBI-G	3.1 ± 0.5	63.8 ± 5	66.7 ± 2	N/A^c	N/A^c
DC-PBI-G-240	5.18 ± 0.4	91.5 ± 3	48.5 ± 3	N/A^c	N/A^c

^a Obtained by acid-base titration using 0.1M NaOH; ^b Estimated from the inherent viscosity; ^c The membrane was insoluble in 98 wt% H₂SO₄ and the values could not be tested. The “±” numbers represent the average of at least three testing samples.

Supplementary Figure 14. Through-plane proton conductivity of PA-doped DC-PBI-D-240 and p-PBI-G-240 membranes at varied temperatures.

Supplementary Figure 24. Single-cell performance of PA-doped DC-PBI-G-240 under anhydrous conditions with dry H₂/O₂ at atmospheric pressure and 240 °C (a Pt loading of 0.5 mg/cm² on both sides).

5. Page 8, Table 2: Methods and parameters for calculating molecular weight using intrinsic viscosity should be provided.

Response: Thank you for raising the issue. We have amended the methods and parameters for calculating molecular weight using intrinsic viscosity in the revised manuscript and the changes are marked in BLUE.

In this work, due to the limited solubility of PBI in organic solvents, determining the molecular weight by GPC was not feasible. Instead, we employed inherent viscosity (IV) measurements to estimate the relative molecular weights of PBIs. IVs were determined using a Cannon Ubbelohde viscometer within a temperature-controlled water bath set at 30 °C. The thoroughly neutralized and dried polymers were dissolved in concentrated sulfuric acid (98%).

After measuring the IVs of the polymers, we followed the methodology outlined by Yuan et al. to evaluate the molecular weight. According to Huggins–Kraemer equation, Yuan et al⁷. verified and established a specific linear relationship between log

[IV] and $\log [M_w]$ for PBI solutions in H_2SO_4 . Accordingly, the calculation of IV and M_w was performed using the following equations⁷:

$$\eta = \ln\left(\frac{t}{t_0}\right)/C \quad (1)$$

$$\lg[M_w] = a \lg[IV] + b \quad (2)$$

where C is the concentration of polymer solution (0.2 g/dL), t is the time for the polymer solution to flow through the viscometer, t_0 is the time for concentrated sulfuric acid to flow through the viscometer, and parameter a and b are obtained by the work from Yuan et al. ($a=1.1369$, $b=2.4084$)⁷. The reproducibility in measuring IV was used to assess the relative error for this technique, and was found to be 1%.

Figure. Mark-Houwink plot for PBI in 96% sulfuric acid (●) and in DMF-5% LiCl (○).

6. Page 10, Line 230-235: (i) The P/N ratio obtained from XPS and FT-IR spectra proved that the PA bridge in the membrane was increased after branched chain modification and 240 °C heat treatment, but the hydrolysis loss of PA bridge in DC-PBI-G-240 membrane after soaking in KOH was the least. This opposite phenomenon needs to be explained carefully. (ii) Supplementary Figure 3: the TGA curves of all membranes should be supplemented, and the process and causes of thermal weight loss should be described in detail. It can be seen from the TGA curve that the membrane starts to lose weight at about 210 °C, so the thermal stability of the membrane is difficult to ensure stable operation at 240 °C.

Response: Thank you for bringing up these issues.

(i) The P/N ratio obtained from XPS and FT-IR spectra proved that the PA bridge in the membrane was increased after branched chain modification and 240 °C heat treatment, but the hydrolysis loss of PA bridge in DC-PBI-G-240 membrane after soaking in KOH was the least, indicating that not all the phosphate bridges in DC-PBI-G-240 had been broken in the KOH treatment. The weight loss of the membranes studied in this work after impregnation with KOH can be attributed to two main factors. Firstly, the degradation of the imidazole rings in the PBI backbone due to the attack of OH⁻ significantly contributed to the decline in membrane weight, resulting in a weight loss of 25% to 30% in m-PBI-D and p-PBI-G. Additionally, about 10% more weight loss occurred in OH-PBI-G due to the hydrolysis of the phosphate bridges. 20% less weight loss from DC-PBI-G and DC-PBI-G-240 membranes than those of HO-PBI-G and HO-PBI-G-240 was observed due to increasing resistance of OH⁻ attacks from the double cross-linked networks.

Figure. Schematic illustration depicting the two degradation routes of PBI under alkaline conditions.

(ii) TGA curves of all membranes had been supplemented in Supplementary Fig. 10. The initial weight loss around 200 °C for each PBI membrane could be attributed to the elimination of residual moisture and solvent^{8,9}. The subsequent larger weight loss observed around 600 °C corresponded to the decomposition of the main polymer chains.

Notably, all membranes exhibited only a 1% weight loss at 300 °C, supporting remarkable thermal stability in stable FC operation up to 240 °C.

Supplementary Figure 10. TGA curves of PA-undoped PBI membranes.

7. Page 11, Table 3: High humidity during fuel cell operation is the main reason for the loss of PA from PEM. Therefore, the relative humidity should be considered in the process of ADL retention test.

Response: Thank you for raising this issue. Relative humidity (RH) has been considered in the process of ADL retention test. As shown in Supplementary Fig. 12, DC-PBI-G demonstrated a significantly improved ADL retention rate of 77.7% even under high humidity of 40% RH at 80 °C, highlighting its effectiveness in mitigating PA loss at lower temperatures and humidified conditions.

Supplementary Figure 12. (a) The ADL values and (b) ADL retention of PA-doped membranes at 80 °C/40% RH.

8. Page 13, Fig. 4: As the proton conductivity test results of m-PBI-D, the operating temperature of PA-PBI membrane is limited below 200 °C due to the dehydration polymerization of PA. However, the proton conduction of p-PBI-G membrane prepared by sol-gel method can be realized above 200 °C, and the dehydration polymerization amount of PA in the membrane should be investigated above 200 °C. The reason for the better proton conduction stability of DC-PBI-G membrane at 240 °C should be elaborated in detail. The data in Fig. 4b and 4c are duplicated, and one of them is retained.

Response: Thank you for raising the issue regarding the dehydration polymerization of PA in the membranes at 240 °C. To investigate such condensation behavior of PA molecules, we analyzed the ³¹P NMR spectra of PA solutions with concentrations ranging from 85% to 115%. Three distinct peaks were observed for orthophosphoric acid (P₁), pyrophosphoric acid (P₂), and triphosphoric acid or polyphosphoric acid (P₃). Solid-state ³¹P NMR spectra of DC-PBI-G-240 (Fig. 4b) exhibited the least compositions (only 4%) of dehydrated products P₂ and P₃, while those of PA-doped p-PBI-G-240 and p-PBI-G-240 showed much higher compositions (44%) for dehydration products of P₃ and P₂ respectively. These results indicated that PA molecules could be preserved better in the double cross-linked structures of DC-PBI-G, leading to better proton conduction stability.

To conclude, the superior proton conductivity stability of DC-PBI-G at 240 °C was attributed to its ability to effectively anchor and retain a high level of PA, coupled with its capacity to mitigate PA dehydration and condensation, and remarkable resistance to creep. Consequently, the synergistic effects, arising from the double cross-linked structure of the DC-PBI-G membrane, substantially enhanced its superior proton conductivity stability under high temperatures above 200 °C.

Moreover, we utilize Fig. 5b to emphasize DC-PBI-G's exceptional absolute values of proton conductivity after extended operation, while Fig. 5c illustrated the variation in through-plane proton conductivity retention over time, offering a more intuitive comparison for proton conductivity retention. Thus, we believe retaining both figures is valuable for providing comprehensive insights into our work.

Fig. 4 | ^{31}P NMR spectra. (a) Liquid-state ^{31}P NMR spectra of 85% PA solution (85PA), 105% PA solution (105PA) and 115% PA solution (115PA or PPA), externally referenced to H_3PO_4 . (b) Solid-state ^{31}P NMR spectra of PA-doped DC-PBI-G-240, p-PBI-G-240 and m-PBI-D-240 membranes, externally referenced to $(\text{NH}_4)_2\text{HPO}_4$.

Supplementary Table 4. The relative integrated peak intensities of ^{31}P NMR spectra.

Samples	P ₁	P ₂	P ₃
115PA (PPA)	5	63	32

105PA	53	45	2
85PA	100	0	0
PA-doped p-PBI-G-240	56	2	42
PA-doped m-PBI-D-240	56	42	2
PA-doped DC-PBI-G-240	96	2	2

Reviewer #2 (Remarks to the Author):

The manuscript by Xue and coworkers reported a novel double cross-linked polybenzimidazole (PBI) membranes for high temperature proton exchange membrane fuel cells (HT-PEMFCs) to operate above 200 °C. The first cross-link comes from tri-functional monomers, and the second cross-link comes from phenol-based phosphoric acid condensation. The two types of interactions were clearly elucidated by comparing with the corresponding control samples. By combining with the sol-gel membrane forming techniques, the resultant PBI membranes show one of the highest ionic conductivity and HT-PEMFC peak power density among literature. More importantly, the authors show stable HT-PEMFC performance at 220 °C. I think this manuscript is suitable to be published in Nature Communications after making some minor revisions.

1. On lines 160-162, the authors claimed that “Additional cross-linking caused by the branching PBI chains had further expanded d1 peak spacing to 9.5 Å in DC-PBI-G PEMs, resulting in larger free pore volumes for higher acid doping levels (ADL).” It seems the authors correlated the physical meaning of WAXD d1 and d2 spacing with the free pore volume, which is a little confusing. From Fig. 1 f-h, the inter-layer spacing should be much larger than the ~9.5 Å length scale. May the authors explain this in more detail?

Response: Thanks for raising the issue. We correlated the physical meaning of d-spacing in WAXD with the packing efficiency and spatial configuration of PBI molecular chains. The WAXD analysis illustrated in Fig. 1i-l revealed that the dense m-PBI-D membrane displayed a singular diffraction peak centered at 3.8 Å, attributed to the π - π stacking of PBI chains. The PPA sol-gel process allowed PBI polymeric chains to interact freely with PPA molecules in the sol state, mitigating the tight packing of polymer segments and leading to the enlargement of intermolecular spaces. Thus, a distinctive XRD peak with d₁ spacing range from 8.4-9.5 Å, alongside the original m-PBI-D peak at 3.8 Å, was observed for all the membranes obtained from the unique

PPA sol-gel process. The 3D model of DC-PBI-G structure after PA removal, generated through atomistic molecular dynamics (PA) simulation, was in good agreement with the WAXD analysis, suggesting the presence of larger intermolecular spaces and free volume for PA storage.

Supplementary Figure 7. 3D spatial configuration and 2D local enlarged views of DC-PBI molecular chains (cubic box size: $12 \times 12 \times 12 \text{ nm}^3$) in (a) PPA, (b) PA, and (c) after PA removal.

It's worth noting that sub-nanoscale intermolecular distance discussed here, with d-spacing of about 3-10 Å from WAXD analyses, should not be confused with the nanoscale layers observed in SEM images of Fig. 1 f-h with a thickness of approximately 30 nm. The latter should be the assemblies of lots of the nanoscale molecular layers. Upon the removal of PA and subsequent freeze-drying, gel-state PBI membranes exhibited split layers composed of loosely assembled nano-filaments. The estimated thickness of these layers, derived from SEM images, approximated 27.9 nm, with the inter-layer spacing measuring around 142.4 nm—a dimension beyond the detection capability of WAXD analysis. To explore the nanometer-level architecture of the membranes, we conducted SAXS analysis, and the findings are detailed in

Supplementary Fig. 6. The 2D SAXS images in Supplementary Fig. 6a further validated the oriented alignment of PBI nano-layers, thereby affirming their anisotropic nature within 3D layered structures¹⁰. Supplementary Fig. 6b revealed distinct peaks at d-spacings of 13.9 nm, 4.7 nm, and 3.3 nm for DC-PBI-G membrane. These peaks may correspond to the filament bundle sizes of 15 nm or the inter-filament distances.

Supplementary Figure 6. (a) 2D SAXS images and (b) SAXS patterns of the undoped DC-PBI-G membrane in both in-plane and trans-plane directions. Quantitative analysis includes (c) layer thickness, (d) inter-layer spacing, and (e) diameter of nano-filaments, derived from SEM images.

2. The stacked FT-IR spectra in Fig. 2b-c and the explanation of the potential P-O-C peak at 1240 cm^{-1} is also confusing. From lines 187-192, the authors claimed “Such P-O-C peak at 1240 cm^{-1} for DC-PBI-G membrane was intensified in DC-PBI-G-240 and weakened in DC-PBI-G-KOH (Fig. 2b)” and “There was no change in the FT-IR spectra of p-PBI-G, p-PBI-G-240 and p-PBI-G-KOH (Fig. 2c)”. However, I see that the 1240 cm^{-1} peak intensity change in Fig. 2b is nearly negligible, while there are obvious changes from $1000\text{-}1250\text{ cm}^{-1}$ in Fig. 2c. Please check and explain this point.

Response: Thank you for raising the issue. We have double check the peak at 1240 cm^{-1} ,

which has been identified as P-O-C according to previous literature¹¹. Supplementary Fig. 8, spanning the range of 1200 to 1350 cm^{-1} , has been included to enhance visibility, clearly demonstrating that the peak intensity increased after thermal treatment at 240 °C and decreased after KOH treatment. Additionally, the observed variations in the 1000-1250 cm^{-1} range in Fig. 2c may be attributed to in-plane bending vibrational modes of imidazole N-H and benzene ring C-H bonds across different treatments on PBIs¹².

Supplementary Figure 8. FT-IR spectra of various PA-undoped membranes.

3. The authors should clarify the “ \pm ” numbers in all the Tables, explaining they are the average of how many samples and whether they are standard deviation numbers.

Response: Thank you for the comment. The ' \pm ' numbers in all the Tables represent the average of at least three testing samples. We appreciate your suggestion, and have clarified this point in the manuscript.

4. The language presentation needs to be polished for this manuscript before publication. For example, there are multiple places showing “ti-functional TMA”, while it should be “tri-functional TMA”; line 55, it should be “lose performance” instead of “lost

performance”. In addition, the authors should unify the use of hyphens, such as “cross-link” or “crosslink”, “FT-IR” or “FTIR”.

Response: Thank you for your comment. Sorry for the errors. We have attempted to address all the issues you pointed out and ensure consistent hyphenation throughout. The changes are marked in BLUE.

Reviewer #3 (Remarks to the Author):

This manuscript reported the PA-doped gel-state PBI membranes with double cross-linked three-dimensional layered structures, prepared via the PPA sol-gel process. The optimized membrane demonstrates excellent proton conductivity, along with successful application in high-temperature fuel cells above 200 °C, yielding impressive power density results. While this work is interesting and suitable for publication in Nature Communications, there are certain aspects that require clarification and modification before acceptance.

1. In reference to the paragraph on lines 134-141, please provide a more detailed explanation of the mechanism behind the formation of the layered structure and its impact on membrane performance. Additionally, elucidate the rationale for investigating this particular aspect of performance.

Response: Thank you so much for raising this issue. The mechanism underlying membrane formation was elucidated through MD simulations and radial distribution function (RDF) calculations, as depicted in Fig. 1m-o and Supplementary Fig. 5. DC-PBI polymer chains exhibited a homogeneous distribution in PPA. However, the hydrolysis of PPA to PA induced a sol-to-gel transition, resulting in noticeable phase separation within the gel-state PA-doped membrane, attributed to the significant solubility difference of PBI in PPA and PA. Subsequent removal of PA revealed a 3D layered structure in the DC-PBI-G membrane. Validation of the oriented alignment of PBI nano-layers was provided by 2D SAXS images in Supplementary Fig. 6a, confirming their anisotropic nature within these 3D layered structures, likely due to rearranged forming stacks in benzimidazole rings parallel to the membrane surface^{13,14}. This unique structure is anticipated to facilitate high acid doping, proton conductivity, and satisfactory gas barrier properties to the present gel-state membrane.

Fig. 1 | Micromorphological characterizations and membrane formation mechanism. SEM surface images of PA-undoped (a) m-PBI-D, (b) p-PBI-G, (c) HO-PBI-G and (d) DC-PBI-G. SEM cross-sectional images of PA-undoped (e) m-PBI-D, (f) p-PBI-G, (g) HO-PBI-G and (h) DC-PBI-G. WAXD spectra of PA-undoped (i) m-PBI-D, (j) p-PBI-G, (k) HO-PBI-G and (l) DC-PBI-G. 3D visualization and spatial configuration of the DC-PBI modelled structure within a $12 \times 12 \times 12$ nm³ amorphous cubic cell, depicted in (m) the presence of PPA, (n) the presence of PA, and (o) post-PA removal.

Supplementary Figure 5. Radial distribution function of DC-PBI/PPA and DC-PBI/PA.

Supplementary Figure 6. (a) 2D SAXS images and (b) SAXS patterns of the undoped DC-PBI-G membrane in both in-plane and trans-plane directions. Quantitative analysis includes (c) layer thickness, (d) inter-layer spacing, and (e) diameter of nano-filaments, derived from SEM images.

2. On lines 18-19, it remains unclear whether the primary objective of the strategies employed in this study is to mitigate PA dehydration and condensation under high-temperature conditions or to prevent PA leaching. We believe further clarification is needed.

Response: Thank you for the question. In response to the inquiry regarding the impact of the double cross-linked structure on mitigating PA condensation or preventing PA leaching, we conducted comprehensive measurements. The results and corresponding discussions have been thoroughly incorporated into both the manuscript and supplementary information.

PA loss significantly impacts PEMFC performance, particularly under harsh operational conditions such as high temperature or high humidity. To provide a more accurate PA retention evaluation, ADL, which combined both ADL_T and ADL_A (Supplementary Fig. 11), was considered in this study. The results in Fig. 3a-b highlighted a notable in-situ PA anchoring process in PA-doped DC-PBI-G. Consequently, its ability to anchor PA and establish extensive 3D hydrogen bonding networks contributed to a much higher ADL retention of 86.0%, retaining 45.4 mol PA/PRU after 50 hours at 240 °C (Fig. 3c-d), compared to p-PBI-G's 18.8 mol PA/PRU and 73.2%. Even under a constant compressive force (equivalent to a stress level of 0.08 MPa) at 240 °C, the ADL retention rate steadily increased from 83.8% for p-PBI-G to 89.5% for HO-PBI-G and 92.6% for DC-PBI-G, highlighting the effectiveness of the double cross-linked structure. Furthermore, DC-PBI-G demonstrated a significantly improved ADL retention rate of 77.7% even under high humidity of 40% RH at 80 °C (Supplementary Fig. 12), indicating its efficient mitigation of PA loss.

Fig. 3 | In-situ anchoring of PA and ADL retention of PA-doped membranes at 240 °C. The changes in P/N ratio and ADL_A over time for (a) DC-PBI-G and (b) p-PBI-G. (c) ADL values for both membranes. (d) Comparison of ADL retention between DC-PBI-G and p-PBI-G.

Supplementary Figure 11. The changes in ADL_T and ADL_A over time at 240 °C for PA-doped (a) DC-PBI-G, (b) p-PBI-G.

Supplementary Figure 12. (a) The ADL values and (b) ADL retention of PA-doped membranes at 80 °C/40% RH.

To investigate the condensation behavior of PA molecules, we analyzed the liquid-state ^{31}P NMR spectra of PA solutions with concentrations ranging from 85% to 115%. These spectra (Fig. 4a), consistent with existing literature⁴, revealed three distinct peaks attributed to orthophosphoric acid (P_1), pyrophosphoric acid (P_2), and triphosphoric acid or polyphosphoric acid (P_3)¹. The solid-state ^{31}P NMR of m-PBI-D-240 and p-PBI-G-240 showed more than 44% of PA had been dehydrated to form P_2 or P_3 after high temperature treatment, while ^{31}P NMR spectra of DC-PBI-G-240 membrane suggested that 96% of the PA molecules (Fig. 4b) had been safely retained without further dehydration or condensation (Supplementary Table 4). These results indicated that PA molecules could be preserved better in the double cross-linked structures of DC-PBI-G, leading to better proton conduction stability.

Overall, on the basis of above results, the formation of double cross-linked structure in this work not only enhanced acid retention levels but also mitigated PA dehydration and condensation under high operating temperatures.

Fig. 4 | ^{31}P NMR spectra. (a) Liquid-state ^{31}P NMR spectra of 85% PA solution (85PA), 105% PA solution (105PA) and 115% PA solution (115PA or PPA), externally referenced to H_3PO_4 (liquid). (b) Solid-state ^{31}P NMR spectra of PA-doped DC-PBI-G-240, p-PBI-G-240 and m-PBI-D-240 membranes, externally referenced to $(\text{NH}_4)_2\text{HPO}_4$ (solid).

Supplementary Table 4. The relative integrated peak intensities of ^{31}P NMR spectra.

Samples	P ₁	P ₂	P ₃
115PA (PPA)	5	63	32
105PA	53	45	2
85PA	100	0	0
PA-doped p-PBI-G-240	56	2	42
PA-doped m-PBI-D-240	56	42	2
PA-doped DC-PBI-G-240	96	2	2

3. If the aim is to mitigate PA condensation at high temperatures, consider incorporating phosphorus NMR spectra for validation. If the emphasis is on preventing PA leaching, please include additional data on low-temperature FC performance and the shut-down/start-up cycling operations.

Response: Thank you for the reviewer's insightful suggestions. We have revised the manuscript accordingly and the changes are marked in BLUE.

To investigate the PA dehydration and condensation at high temperatures, we had used ^{31}P NMR spectra to study the PA solutions and the PA doped membranes based on the three distinct peaks corresponding to orthophosphoric acid (P_1), pyrophosphoric acid (P_2), and triphosphoric acid or polyphosphoric acid (P_3), as shown in Fig. 4. As a result, PA-doped DC-PBI-G-240 exhibited the highest relative integrated peak intensity of orthophosphoric acid (P_1), suggesting its substantial retention of PA molecules without further dehydration or condensation.

Fig. 4 | ^{31}P NMR spectra. (a) Liquid-state ^{31}P NMR spectra of 85% PA solution (85PA), 105% PA solution (105PA) and 115% PA solution (115PA or PPA), externally

referenced to H_3PO_4 . (b) Solid-state ^{31}P NMR spectra of PA-doped DC-PBI-G-240, p-PBI-G-240 and m-PBI-D-240 membranes, externally referenced to $(\text{NH}_4)_2\text{HPO}_4$.

Supplementary Table 4. The relative integrated peak intensities of ^{31}P NMR spectra.

Samples	P ₁	P ₂	P ₃
115PA (PPA)	5	63	32
105PA	53	45	2
85PA	100	0	0
PA-doped p-PBI-G-240	56	2	42
PA-doped m-PBI-D-240	56	42	2
PA-doped DC-PBI-G-240	96	2	2

Furthermore, the PA retention capability of DC-PBI-G membranes under lower temperature was also evaluated via polarization curves from 25 °C to 80 °C, long-term durability, and shut-down/start-up cycling tests. The results demonstrated that DC-PBI-G MEA exhibited excellent output performance (443 mW/cm², as shown in Supplementary Fig. 22) and sustained operation for over 170 hours at low temperatures (e.g., 40 °C), with a minimal decay rate of 8.97 μV/h. After a more demanding accelerated stress test (AST) involving 100 start-up/shut-down cycles, it maintained a peak power density of 331 mW/cm², corresponding to a 75% retention (Supplementary Fig. 23). The DC-PBI-G membrane demonstrated superior FC performance and operational flexibility under both low-temperature and high-temperature conditions (Fig. 6f and Supplementary Table 7-9).

Supplementary Figure 22. Low-temperature single cell performance of PA-doped DC-PBI-G membranes with dry H₂ and O₂ at atmospheric pressure without additional humidification (a Pt loading of 1.0 mg/cm² on both sides).

Supplementary Figure 23. (a) Long-term durability of PA-doped DC-PBI-G MEA with H₂/O₂ at 0.2 A/cm² and 40 °C. (b) Peak power density values of the PA-doped DC-PBI-G MEAs after the shut-down/start-up AST cycling at 40 °C. Testing conditions: a Pt loading of 1.0 mg/cm² on both sides, dry H₂/O₂ flow, without backpressure.

4. Please elucidate the influence of high proton conductivity on device performance and the potential reduction in Pt loading. If the study covers not only high-temperature aspects, we recommend evaluating low-temperature durability as well. Additionally, the current 100-hour durability test in Fig.5e is inadequate to meet contemporary HT-PEMFC requirements.

Response: Thank you for bringing up these important aspects. We have revised the manuscript accordingly and the changes are marked in BLUE.

In practical fuel cell applications, the proton exchange membrane (PEM) is crucial for proton conduction and gas separation. High proton conductivity in the PEM could improve fuel cell efficiency and allow for lower precious metal catalyst loading. Despite reducing the Pt loading from 1.0 mg/cm² to 0.5 mg/cm², the superior proton conductivity of DC-PBI-G still achieved a peak power density of 886 mW/cm² at 200 °C. This underscored its capability for high-performance output even with low catalyst loading.

Supplementary Figure 17. Single-cell performance of PA-doped DC-PBI-G under anhydrous conditions with dry H₂/O₂ at atmospheric pressure (a Pt loading of 0.5 mg/cm² on both sides).

Thank you for the reviewer's valuable suggestion regarding FC durability. The high-temperature durability testing period was extended to 255 hours (Fig. 6e) utilizing m-PBI-D, DC-PBI-G and p-PBI-G MEAs at 220 °C and 0.2 A/cm² current load without external humidification. p-PBI-G showed significant voltage decay rate of 0.53 mV/h and severe high-temperature creep with observable cracks and holes in the MEA (Supplementary Fig. 21), leading to operational failure at 150 hours. In contrast, the DC-PBI-G based fuel cell demonstrated a substantially lower voltage decay rate of only 0.27 mV/h over a 250-hour testing period, attributing to the presence of phosphate bridges and a double cross-linked structure. Additionally, low-temperature durability tests were conducted at 40 °C and the results were included in Supplementary Fig. 23a. DC-PBI-G MEA also sustained stable operation over 170 hours with only a low voltage decay rate of 8.97 μV/h, assigned to its robust PA retention capability even under humidified conditions.

Fig. 6 | Single-cell performance of PA-doped membranes under anhydrous conditions with dry H₂/O₂ at atmospheric pressure (a Pt loading of 1.0 mg/cm² on both sides). (a) Polarization curves at different temperatures. (b) Power density curves at different temperatures. Single cell performance comparison based on DC-PBI-G, p-PBI-G and m-PBI-D membranes at (c) 160 °C and (d) 200 °C. (e) Long-term durability test at 0.2 A/cm² and 220 °C. (f) Comparison of the peak power density of high-temperature (HT)-PEMFCs.

Supplementary Figure 21. Photographs of the PA-doped (a) DC-PBI-G and (b) p-PBI-G-based MEAs after a long-term durability test at 0.2 A/cm^2 and $220 \text{ }^\circ\text{C}$ with dry H_2/O_2 under atmospheric pressure (a Pt loading of 1.0 mg/cm^2 on both sides).

Supplementary Figure 23. (a) Long-term durability of PA-doped DC-PBI-G MEA with H₂/O₂ at 0.2 A/cm² and 40 °C. (b) Peak power density values of the PA-doped DC-PBI-G MEAs after the shut-down/start-up AST cycling at 40 °C. Testing conditions: a Pt loading of 1.0 mg/cm² on both sides, dry H₂/O₂ flow, without backpressure.

5. Regarding the layered microstructure of the membranes, please provide a comparison of proton conductivity along both the in-plane and through-plane directions.

Response: Thank you for raising the issue. The in-plane and through-plane proton conductivity of DC-PBI-G and p-PBI-G were compared in Supplementary Fig. 13, while anisotropic proton transport pathways were illustrated in Supplementary Fig. 13a. For PA-doped DC-PBI-G and p-PBI-G membranes, the layered micro-structure increased proton transfer tortuosity along the through-plane direction, resulting in slightly lower through-plane conductivity.

Supplementary Figure 13. (a) Schematic diagram of proton transport pathways along the in-plane and through-plane directions. (b) Comparison of in-plane and through-plane proton conductivity for DC-PBI-G. (c) Comparison of in-plane and through-plane proton conductivity for p-PBI-G.

6. The language in the abstract needs to be polished to enhance the emphasis on the key findings and core contributions of the work.

Response: We appreciate your valuable suggestion. The language in the abstract has been refined and polished accordingly. The changes are marked in BLUE.

Reference

- 1 Sannigrahi, P. & Ingall, E. Polyphosphates as a source of enhanced P fluxes in marine sediments overlain by anoxic waters: Evidence from ^{31}P NMR. *Geochemical Transactions* **6**, 52-59 (2005).
- 2 Perry, K. A. *et al.* A comparative study of phosphoric acid-doped m-PBI membranes. *J. Polym. Sci. Pt. B-Polym. Phys.* **52**, 26-35 (2014).
- 3 Pingitore, A. T., Molle, M., Schmidt, T. J. & Benicewicz, B. C. in *Encyclopedia of Sustainability Science and Technology* (ed Robert A. Meyers) 1-38 (Springer New York, 2018).
- 4 Aihara, Y., Sonai, A., Hattori, M. & Hayamizu, K. Ion conduction mechanisms and thermal properties of hydrated and anhydrous phosphoric acids studied with ^1H , ^2H , and ^{31}P NMR. *J. Phys. Chem. B* **110**, 24999-25006 (2006).
- 5 Kutagulla, S. *et al.* Comparative studies of atomically thin proton conductive films to reduce crossover in hydrogen fuel cells. *ACS Appl. Mater. Interfaces* **15**, 59358-59369 (2023).
- 6 Neyerlin, K. C., Singh, A. & Chu, D. Kinetic characterization of a Pt-Ni/C catalyst with a phosphoric acid doped PBI membrane in a proton exchange membrane fuel cell. *J. Power Sources* **176**, 112-117 (2008).
- 7 Yuan, Y., Johnson, F. & Cabasso, I. Polybenzimidazole (PBI) molecular weight and Mark-Houwink equation. *Journal of Applied Polymer Science* **112**, 3436-3441 (2009).
- 8 Harilal, Nayak, R., Ghosh, P. C. & Jana, T. Cross-linked polybenzimidazole membrane for PEM fuel cells. *ACS Appl. Polym. Mater.* **2**, 3161-3170 (2020).
- 9 Zhu, T. *et al.* Gel-state polybenzimidazole proton exchange membranes with flexible alkyl sulfonic acid side chains for a wider operating temperature range (25-240 °C). *J. Energy Chem.* **85**, 91-101 (2023).
- 10 Wang, X. *et al.* Mechanical nonreciprocity in a uniform composite material. *Science* **380**, 192-198 (2023).
- 11 Yu, S. & Benicewicz, B. C. Synthesis and properties of functionalized polybenzimidazoles for high-temperature PEMFCs. *Macromolecules* **42**, 8640-8648 (2009).
- 12 Dai, Q. *et al.* High-performance PBI membranes for flow batteries: from the transport mechanism to the pilot plant. *Energy Environ. Sci.* **15**, 1594-1600 (2022).
- 13 Tang, H. *et al.* Properties and stability of quaternary ammonium-biphosphate ion-pair poly(sulfone)s high temperature proton exchange membranes for H_2/O_2 fuel cells. *J. Power Sources* **475**, 228521 (2020).
- 14 He, R., Li, Q., Bach, A., Jensen, J. O. & Bjerrum, N. J. Physicochemical properties of phosphoric acid doped polybenzimidazole membranes for fuel cells. *J. Membr. Sci.* **277**, 38-45 (2006).

REVIEWERS' COMMENTS

Reviewer #1 (Remarks to the Author):

All of my concerns have been addressed. Hence, I think it can be considered and accepted by Nature Communications.

Reviewer #2 (Remarks to the Author):

The authors have addressed the reviewers' comments in detail and the manuscript has been thoroughly revised. I recommend the manuscript to be published in Nature Communications as it is.

Reviewer #3 (Remarks to the Author):

The manuscript has been well-revised according to the reviewer's comments and could be published now in NC.